# A viral expression factor behaves as a prion

Hao Nan [1], Hongying Chen[1], Mick F. Tuite[2] & Xiaodong Xu [1]

Prions are proteins that can fold into multiple conformations some of which are self-propagating. Such prion-forming proteins have been found in animal, plant, fungal and bacterial species, but have not yet been identified in viruses. Here we report that LEF-10, a baculovirus-encoded protein, behaves as a prion. Full-length LEF-10 or its candidate prion-forming domain (cPrD) can functionally replace the PrD of Sup35, a widely studied prion-forming protein from yeast, displaying a [$PSI^+$]-like phenotype. Furthermore, we observe that high multiplicity of infection can induce the conversion of LEF-10 into an aggregated state in virus-infected cells, resulting in the inhibition of viral late gene expression. Our findings extend the knowledge of current prion proteins from cellular organisms to non-cellular life forms and provide evidence to support the hypothesis that prion-forming proteins are a widespread phenomenon in nature.

---

[1] State Key Laboratory of Crop Stress Biology for Arid Areas and College of Life Sciences, Northwest A&F University, 712100 Yangling, Shaanxi, China. [2] Kent Fungal Group, School of Biosciences, University of Kent, Canterbury CT2 7NJ, UK. Correspondence and requests for materials should be addressed to X.X. (email: xuxd@nwsuaf.edu.cn)

P rions are proteins that can take up multiple conformations that show distinct aggregation propensities and where at least one of the conformations is self-propagating. In mammals, the prion protein PrP can induce the formation of amyloid plaques in the central nervous system and cause neurodegenerative diseases[1,2]. Such prion diseases can affect diverse species, e.g., kuru, Creutzfeldt–Jacob disease, fatal familial insomnia, and Gerstmann-Sträussler-Scheinker disease in humans, bovine spongiform encephalopathy in cattle, scrapie in sheep and goats, and chronic wasting disease in deer and elk. Sharing common characteristics with PrP, a number of prion-forming proteins have been identified in yeast and filamentous fungi: [PSI+][3], [URE3][3], [PIN+] or also known as [RNQ+][4], [SWI+][5], [OCT+][6], [MOT3+][7], [MOD+][8], and [Het-s][9]. Fungal prions are not typically detrimental to their hosts[10], but they are useful in helping us to understand how mammalian prions are propagated. Recently, prion-forming proteins were discovered in plant[11] and bacteria[12,13]. Most non-mammalian prions, if not all, serve as modulators of non-pathogenic, epigenetic phenotypes and in several cases they may actually play important roles in normal biological processes, providing growth advantages under certain conditions[14–16]. For example, Luminidependens (LD), the first identified plant protein behaving as a prion, is a transcriptional regulator involved in the vernalization[11,17]. Such a self-propagating conformational change in LD is considered to be a signal that can switch plants between reproductive and vegetative growth[11]. This suggests that prion-like conversion can act as a sensor, responding to environmental change, and subsequently regulate gene expression and cellular processes.

It has long been recognized that prion-based mechanisms provide a plausible regulatory pathway for a host's response to environmental stress. The rate of induction of the [PSI+] prion in yeast, where it acts as an epigenetic modifier of translation termination, increases when cells are exposed to oxidative stress or high salt conditions[18]. The yeast prion [MOT3+], which regulates the expression of cell wall biogenesis-related genes, enables the cell to tolerate high concentrations of ethanol[19]. [MOD+], a prion state of the yeast tRNA isopentenyl transferase, contributes to the resistance of yeast cell to antifungal drugs such as fluconazole and ketoconazole[8]. Also, prion-like conformation of mitochondrial antiviral signaling protein (MAVS) and the apoptosis-associated protein ASC in mammalian cells activate the downstream immunological or inflammatory signaling pathways[20,21]. These findings suggest that the ability to switch a protein to a heritable prion form under certain conditions may offer the host evolutionary advantages.

The potential evolutionary advantage suggests that prion proteins should exist widely in various types of organisms. Viruses are the most abundant form of life on earth. However, a virus-encoded prion has not been experimentally identified, although prion-like domains in phage have been predicted by bioinformatic methods[22]. Autographa californica multiple nucleopolyhedrovirus (AcMNPV) is a large double-stranded DNA virus that infects insects. It is the archetype species of the genus Alphabaculovirus of the family Baculoviridae, and is widely used as a vector for the production of recombinant proteins in the pharmaceutical industry. The AcMNPV-encoded LEF-10 protein has been identified as a viral late expression factor[23] and is thought to be one of the genes positively selected for during baculovirus evolution[24]. Although LEF-10 in Bombyx mori nucleopolyhedrovirus (BmNPV) and AcMNPV have been identified as being essential for the viral DNA replication and gene expression[25,26], the exact role of LEF-10 in virus infection has not been extensively studied. We have previously reported that LEF-10 can form unique detergent (SDS)-resistant, high molecular weight aggregates[27] and

that the aggregation of LEF-10 can be induced by over-production leading to subsequent down-regulation of viral late gene expression[25]. These characteristics of the LEF-10 protein mirror those of a prion protein. In this study, we have investigated the prion behavior of LEF-10 in a heterologous (yeast) system and in virus-infected insect cells.

## Results

**Inducible aggregation of LEF-10 in infected cells**. To investigate the aggregation behavior of LEF-10 under physiological conditions, we developed a recombinant baculovirus sensor system transmitting the aggregation state of LEF-10 to a downstream gene output. The sensor system comprises a LEF-10-EGFP driven by the native lef-10 promoter and an mCherry reporter regulated by a baculovirus very late promoter (the p10 promoter) (Fig. 1a and Supplementary Fig. 1e). In living baculovirus-infected Spodoptera frugiperda (Sf9) cells, LEF-10-EGFP shows two distinct distribution patterns, diffused or aggregated. The expression levels of mCherry were high in cells with evenly diffused LEF-10-EGFP, but the fluorescence of mCherry was undetectable in the cells containing LEF-10-EGFP aggregates[25] (Fig. 1b). Further computational analysis showed that once such aggregation occurred, most of the LEF-10 protein concentrated in small areas in the infected cells (Fig. 1c). In our designed genetic sensor system, correlation of the LEF-10-EGFP state with its regulated mCherry readout implied that aggregation of LEF-10 limited the availability of the fusion protein to activate the AcMNPV late promoter thereby reducing the expression of the mCherry reporter. We speculate that aggregated nuclei of LEF-10-EGFP could be inducing soluble LEF-10-EGFP to adopt the aggregated form, resulting in depletion of soluble, functional LEF-10-EGFP and the subsequent repression of late gene expression. The aggregation propensity of LEF-10-EGFP in infected cells is consistent with the behavior described for previously identified prions[28,29].

**LEF-10 behaves as a prion in a yeast prion reporter assay**. In order to verify the prion characteristics of LEF-10, we employed an assay based on the well-characterized prion phenotypes of the Saccharomyces cerevisiae translation termination factor Sup35[7]. This protein consists of an N-terminal modular prion-forming domain (PrD), a highly charged middle region (M) and a C-terminal release function domain (C). To determine whether LEF-10 could produce an epigenetic modification of the heterologous functional protein, we substituted the PrD of Sup35 with LEF-10 to generate a LEF-10-Sup35MC fusion protein. Similar to the yeast containing wild-type Sup35, the yeast strain lacking the endogenous SUP35 gene, but harboring LEF-10-Sup35MC exhibited both [PSI+]-like and [psi−]-like phenotypes (Fig. 2a), which were hereafter referred to as [LEF+] and [lef−]. The replacement of Sup35 with the LEF-10-Sup35MC fusion proteins in [lef−] cells gave rise to colonies with a red, Ade− phenotype indicating the LEF-10-Sup35MC fusion protein could efficiently terminate translation at the premature UGA stop codon in the ade1-14 allele in this strain. In contrast, but similar to [PSI+] strains, [LEF+] cells expressing the chimeric LEF-10-Sup35MC protein showed significantly reduced termination efficiency resulting in the read-through of ade1-14 premature stop codon and the white Ade+ phenotype as seen in [PSI+] cells. This behavior is consistent with the functional sequestration of LEF-10-Sup35MC into an aggregated state in [LEF+] cells leading to reduced translation termination activity.

We further observed that [LEF+] and [lef−] strains could maintain the respective prion or non-prion phenotypes, i.e., suppression or non-suppression of the ade1-14 allele, over many

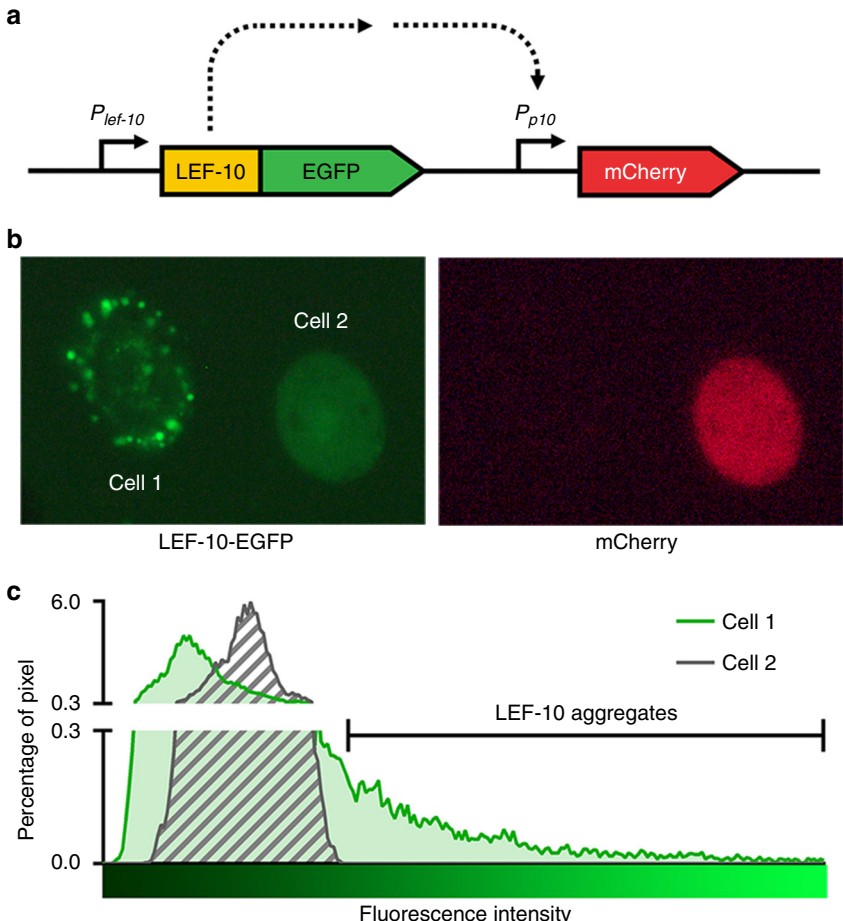

**Fig. 1** LEF-10-EGFP forms aggregates in infected insect cells. **a** Diagram of the recombinant baculovirus sensor system functions by transmitting the aggregation state of LEF-10-EGFP to downstream gene reporter mCherry outputs. The cassette fragment was integrated into the *polh* locus of a *lef-10* knock-out Bacmid via site-specific homologous recombination (Supplementary Fig. 1a). **b** LEF-10-EGFP exists in two distinct forms: as aggregated (Cell 1) or as diffuse (Cell 2) forms in infected insect cells (MOI = 3) at 60 hpi (left). The mCherry reporter protein representing late gene expression can only be detected in cells harboring diffuse LEF-10-EGFP (right). **c** The distribution of LEF-10-EGFP in the two cells (b, left) was analyzed by ImageJ software. The proportion of the pixels of a certain brightness to all the pixels harbored by one cell is defined as "percentage of pixel". Higher fluorescence intensity, which leads to the curve shifting to the right in Cell 1, indicates the aggregation of LEF-10-EGFP and exhausts the pool of non-aggregated LEF-10-EGFP which occurs in the lower fluorescence intensity areas

cell generations. The white Ade$^+$ phenotype was stable during the propagation of [*LEF*$^+$] colonies, while [*lef*$^-$] strain spontaneously gave rise to [*LEF*$^+$] colonies at a very low frequency (Fig. 2a). As the *SUP35* native promoter was used for the expression of the LEF-10-Sup35MC fusion protein, this observation suggested that the prion conformation of LEF-10 was self-perpetuating and the low expression level was sufficient for the maintenance of its prion state.

Most of prion proteins have the ability to form SDS-resistant polymers. The SDS resistance of protein complexes on semi-denaturing detergent agarose gels (SDD-AGE) can distinguish highly ordered amyloid fibrils from disordered superstructures[30]. To assess whether LEF-10 possesses this typical SDS-resistant characteristic, we examined the LEF-10-Sup35MC fusion protein in [*LEF*$^+$] and [*lef*$^-$] strains using SDD-AGE. Consistent with the in vivo phenotypes, the LEF-10-Sup35MC protein formed high-molecular-weight, SDS-resistant aggregates in [*LEF*$^+$] yeast cells, but could only be detected as monomers in [*lef*$^-$] yeast cells (Fig. 2b). To exclude the possibility that different protein levels contributed to the formation of aggregation bands, the expression levels of chimeric LEF-10-Sup35MC proteins were tested in parallel by Western blot using SDS-PAGE (Fig. 2b, the bottom two panels). Detection of similar expression levels of these

chimeric proteins suggested that the SDS-resistant characteristic harboring by LEF-10 was its innate biochemical property.

As a molecular chaperone, Hsp104 is indispensable for the heritability of yeast amyloid prions[31]. To test whether Hsp104 is essential for the maintenance of the [*LEF*$^+$] phenotype, the chaperone function of Hsp104 was inactivated using guanidine hydrochloride (GdnHCl), an inhibitor of the ATPase activity of Hsp104[32]. By growing [*LEF*$^+$] cells on medium containing 5 mM GdnHCl, [*LEF*$^+$] strains were restored to [*lef*$^-$] phenotype (Fig. 3a). The dependence of [*LEF*$^+$] on Hsp104 was further investigated by assessing the de novo inducibility of [*LEF*$^+$] in the absence of Hsp104. It was observed that deletion of the non-essential *HSP104* gene eliminated the [*LEF*$^+$] phenotype and these cured [*lef*$^-$] strains only produced red Ade$^-$ colonies (Fig. 3b) indicating that the LEF-10 aggregates required Hsp104 for its propagation and that the [*LEF*$^+$] phenotype in yeast was Hsp104-dependent. Such dependency is found for the majority of yeast prions[31,33] ruling out non-epigenetic effects and demonstrating the *bona fide* prion phenotype of [*LEF*$^+$] strains.

These results support our speculation that LEF-10 is a prion-forming protein that not only can exist in at least one alternative self-perpetuating conformational state, but also was able to switch between a soluble form and a prion-like aggregated form in vivo.

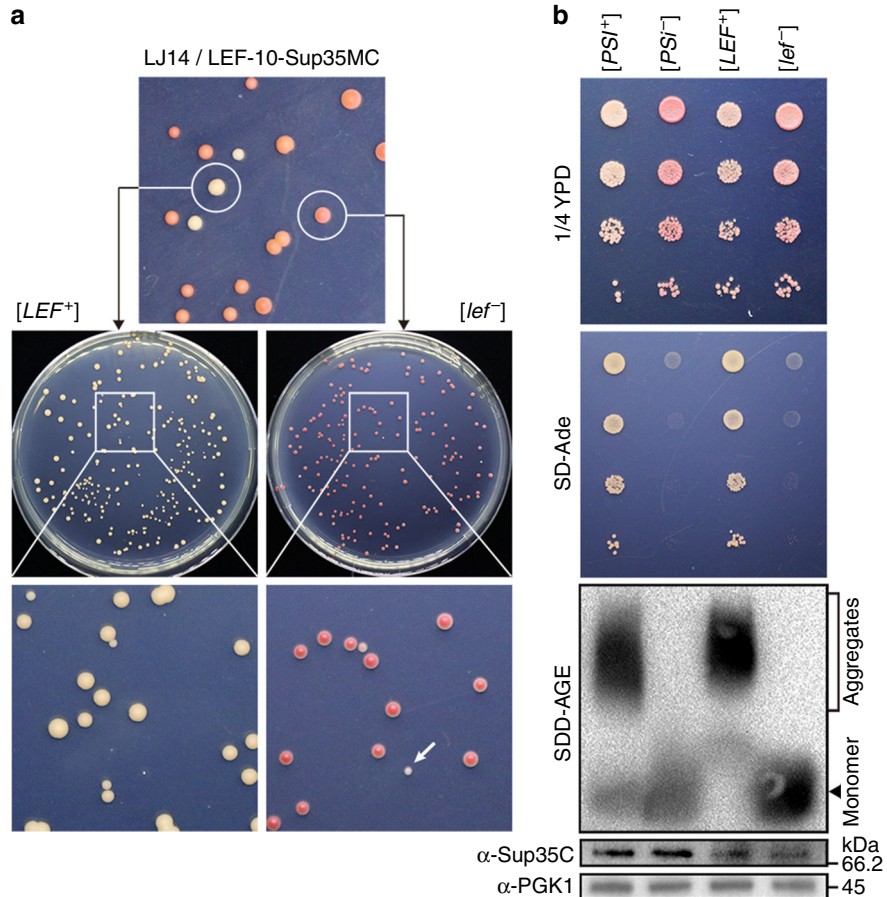

**Fig. 2** LEF-10 functionally replaces the prion domain of the yeast Sup35 protein. **a** Yeast *ade1-14* cells expressing LEF-10-Sup35MC were spread on complete (1/4 YPD) medium. [*LEF+*] strains formed white colonies which were distinguishable from [*lef−*] strains by the red pigment accumulated through the adenine biosynthetic pathway (top). Both phenotypes could be sustained during cell propagation (middle). [*lef−*] strains spontaneously generated [*LEF+*] colonies at a very low frequency (bottom, a colony with the [*LEF+*] phenotype is indicated by an arrow) and these [*LEF+*] colonies displayed Ade+ (suppression of *ade1-14*) phenotype when tested directly on SD-Ade medium. **b** Comparable characteristics of LEF-10-Sup35MC and full-length Sup35 in yeast. Serial dilutions of [*LEF+*] and [*lef−*] strains were spotted on the 1/4 YPD plate (the top panel), or medium lacking adenine (SD-Ade) on which only prion-containing i.e., [*PRION+*] strains suppressing the stop codon in the yeast *ade1-14* allele could grow (the second panel). SDS-resistant aggregates in cell lysates of yeast strains expressing LEF-10-Sup35MC were examined by SDD-AGE (the third panel). The expression levels of full-length Sup35 and LEF-10-Sup35MC were examined by Western blot, probing with a Sup35C-specific antibody (the fourth panel). Endogenous phosphoglycerate kinase 1 (PGK1) was detected with a PGK1-specific antibody and served as a loading control (the bottom panel). [*PSI+*] and [*psi-*] strains were used as positive and negative controls

**Identification of the cPrD in LEF-10**. Almost all identified yeast prion proteins harbor glutamine/asparagine (Q/N)-rich domains, which are responsible for the prion characteristic of self-propagating aggregation. LEF-10 is only 78 amino acids in length and does not contain any Q/N-rich regions. Using PLAAC, an algorithm used to predict PrDs in proteins based on their amino acid sequence[34], we were unable to define a distinct candidate prion-forming domain (cPrD) in LEF-10 (Supplementary Fig. 2a). However, Tango, an algorithm used to predict amyloidogenic regions[35], predicted four adjacent regions in the N-terminal part of LEF-10 as having potential of amyloid formation (Supplementary Fig. 2b). To further define the potential cPrD, we generated three LEF-10 truncations based on the sequence alignment of LEF-10 in 28 baculoviruses[25], with each truncation containing one of the three conserved regions (C1, C2, and C3) of LEF-10 and its flanking sequences (Fig. 4a).

In the Sup35MC-based in vivo assay, LEF-10$_{1-41}$ containing the C1 conserved region maintained [*LEF+*] phenotype when fused to Sup35MC in yeast cells (Fig. 4b, c), consistent with the prediction by the Tango algorithm that a cPrD is located at the N terminus of LEF-10. We next shortened LEF-10$_{1-41}$ from each end and found that a 23 amino acid stretch (LEF-10$_{12-34}$) was able to

drive the [*LEF+*] phenotypes in yeast. Hence, we defined residues 12–34 as the minimal cPrD of LEF-10 (Fig. 4a–c), although this sequence shows no sequence homology with any sequence known to function as a PrD in yeast.

The various LEF-10-Sup35MC fusion proteins tested (namely LEF-10, LEF-10$_{1-41}$ and LEF-10$_{12-34}$) led to the read-through of *ade1-14* premature stop codon in Sup35MC-based assays and the formation of high-molecular-weight fractions detected by SDD-AGE (Fig. 4b). In contrast, those LEF-10-Sup35MC chimeric proteins lacking the cPrD of LEF-10 displayed [*psi−*]-like phenotypes confirming that the [*LEF+*] phenotype observed was dependent on the LEF-10 cPrD (Fig. 4b). Additionally, deletion of the *HSP104* gene eliminated the [*LEF+*]-like phenotypes exhibited by chimeric proteins containing the cPrD of LEF-10 (Fig. 4c), demonstrating that Hsp104 is required for the heritability of [*LEF+*].

**Effects of highly conserved amino acids in the LEF-10 cPrD**. Identification of the LEF-10 cPrD in the most conserved region of C1 prompted us to explore whether the conserved residues in LEF-10 cPrD were associated with the prion-forming property of

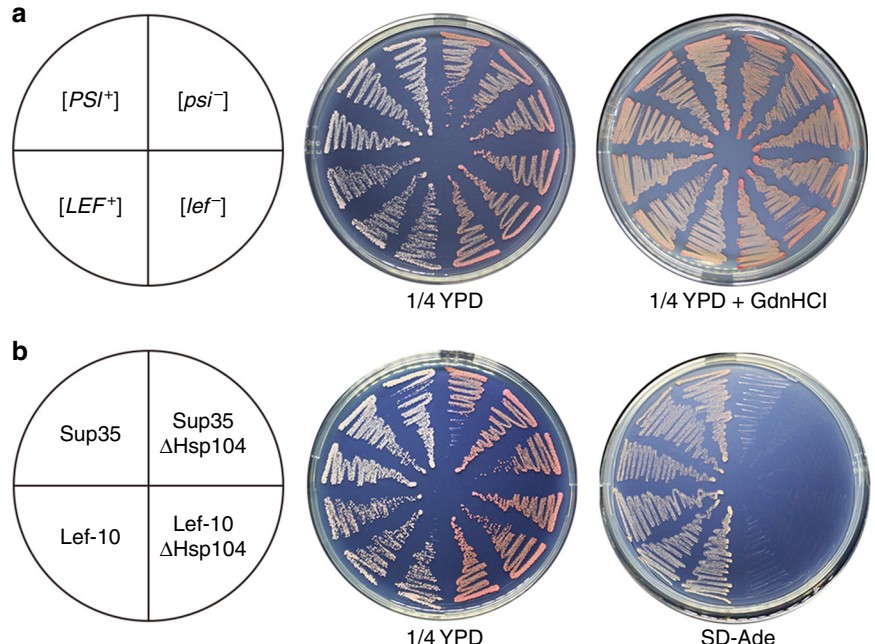

**Fig. 3** The [LEF+] phenotype in yeast is Hsp104-dependent. **a** Phenotypic curing of [LEF+] by guanidine hydrochloride (GdnHCl), an inhibitor of molecular chaperone Hsp104. Phenotypes of [PSI+], [psi−], [LEF+], and [lef−] colonies were, respectively, re-streaked on 1/4 YPD medium with or without 5 mM GdnHCl. **b** The [LEF+] strain phenotype is curable by deletion of the HSP104 gene. The [PSI+] strain was used as control

this region. We therefore systematically substituted each of the ten highly conserved amino acid residues (Fig. 4a) in the identified cPrD with alanine, and tested whether these mutations could eliminate the [LEF+] phenotype in yeast cells. None of these mutations completely eliminated the [LEF+] phenotype or inhibited [LEF+] propagation (Supplementary Fig. 3), suggesting that the primary sequence of the LEF-10 cPrD is not a strict requirement for this property.

We also investigated if the same Ala substitution mutants could replace the wild-type LEF-10 function when placed under the control of a strong promoter and expressed as fusion proteins with a C-terminal EGFP (Supplementary Fig. 1b). Among the mutants, several mutants (LEF-10$^{I17A}$, LEF-10$^{N20A}$, and LEF-10$^{I29A}$) lost the ability to rescue baculovirus using a lef-10 null bacmid and no late gene expression was detected, whereas the late gene expression level regulated by the mutant LEF-10$^{L21A}$ was significantly higher than that of wild-type LEF-10 (Supplementary Fig. 4a and Fig. 5a, b). In order to determine the effect of L21A substitution on the function of LEF-10 under physiological conditions, baculoviruses were further rescued using wild-type LEF-10 or LEF-10$^{L21A}$ expressed under the control of lef-10 native promoter (Supplementary Fig. 1d) and their growth curves were determined. Compared to the virus expressing wild-type LEF-10, the replication speed of the virus rescued by LEF-10$^{L21A}$ decreased and its virus titer plateau was approximately 10-fold lower (Supplementary Fig. 5), suggesting that LEF-10$^{L21A}$ was a functionally down-regulated mutant.

As we had noticed that protein overexpression can induce the aggregation of LEF-10, we postulate that the aggregation tendency of LEF-10$^{L21A}$ may be lower than the wild-type protein, and this property may compensate the down-regulated function of the mutant when overexpressed, and then upregulate its downstream gene expression. Using fluorescence microscopy, we observed that overexpressed LEF-10$^{L21A}$-EGFP formed many fewer aggregates than overexpressed LEF-10-EGFP in baculovirus-infected insect cells (Fig. 5c, d). In transformed yeast cells, LEF-10$^{L21A}$-GFP also produced fewer

and smaller aggregates than LEF-10-GFP (Supplementary Fig. 6). These results are consistent with the L21A mutation reducing the aggregation tendency of LEF-10.

**Prion behavior of LEF-10 in its native host.** To investigate the prion property of LEF-10 in infected cells and the effects of protein aggregation on its function, we defined the two LEF-10 phenotypes in infected cells: LEF-10 existed as small punctate aggregates (LEF$^{Ag}$) in the cytoplasm; LEF-10 was diffused (LEF$^d$) and evenly distributed in the cell and/or localized in the nuclei (Fig. 6b). We counted the cells with the two phenotypes in Sf9 cells expressing LEF-10-EGFP driven by the lef-10 promoter (Supplementary Fig. 1e), and found that the ratio of cells with LEF$^{Ag}$ continuously decreased from 24 to 48 hpi, whereas the ratio of cells with LEF$^{Ag}$ dropped more rapidly in cells expressing LEF-10$^{L21A}$ than those expressing wild-type LEF-10 (Fig. 5e, f). This observation is also consistent with our observation that the L21A mutation reduced the aggregation tendency of LEF-10 even at its normal expression level. SDD-AGE and Western blot analysis showed that the aggregation degree of the L21A mutant is lower than that of the wild type protein (Fig. 5g, h).

Using the p10 promoter driven EGFP as a reporter for baculovirus late gene expression (Supplementary Fig. 1d), we infected insect cells at a multiplicity of infection (MOI) from 0.125 to 64 with the aim of controlling the expression and aggregation of LEF-10. At 36 hpi, we counted the cell numbers expressing EGFP. The results showed that the EGFP positive cells were comparable in LEF-10$^{L21A}$ and wild-type LEF-10 groups at MOI lower than 8, however, at higher MOI, there were obviously more EGFP positive cells in LEF-10$^{L21A}$ group than in the LEF-10 group (Fig. 6a and Supplementary Fig. 7). We also measured the mean fluorescence intensity (MFI) of the EGFP positive cells at 60 hpi, and the data showed that the late gene expression levels were similar in the two groups at an MOI lower than 0.5, but MFIs for the LEF-10$^{L21A}$ expressing cells were much higher than the cells expressing wild-type LEF-10 at an MOI higher than 1

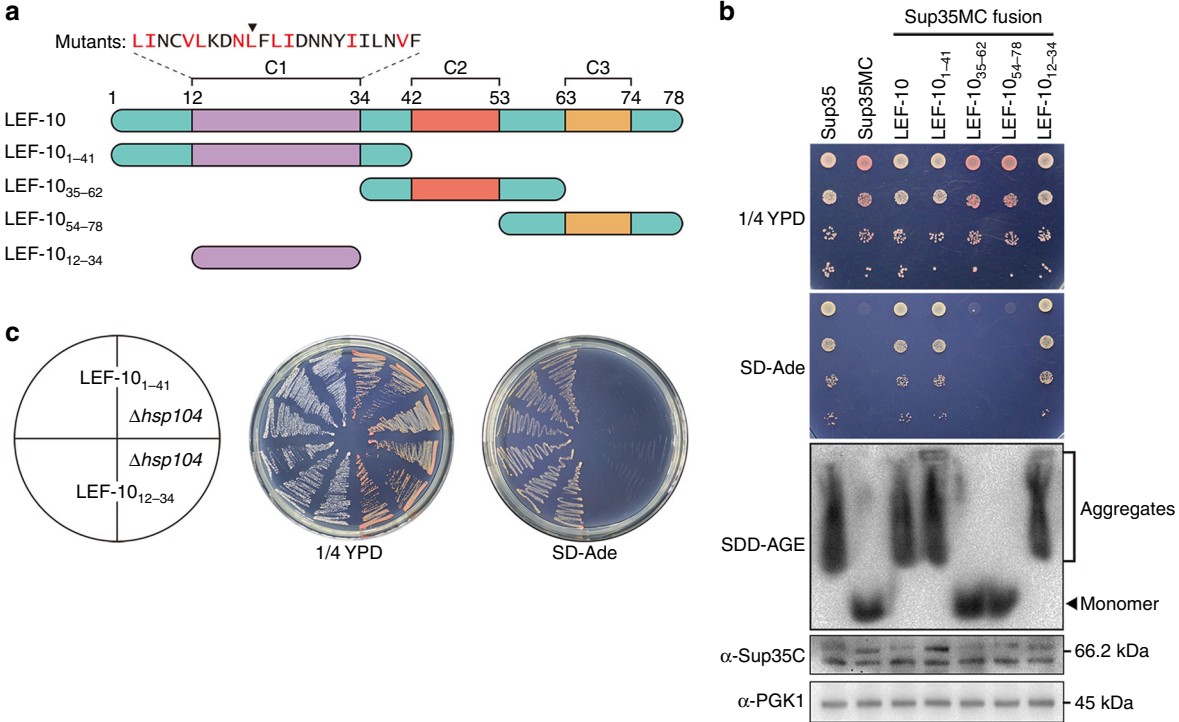

**Fig. 4** Identification of a candidate prion-forming domain (cPrD) in LEF-10. **a** Cartoon of LEF-10 and its truncations. The location of three conserved regions (C1, C2 and C3) and the amino acid sequence of the predicted cPrD are presented above. The ten highly conserved amino acids in C1 are colored in red. L21 is indicated by the inverted triangle. **b** Characterization of LEF-10 truncations in yeast. LEF-10, LEF-10$_{1-41}$ and LEF-10$_{12-34}$ display similar characters to Sup35 on 1/4 YPD and SD-Ade medium, and by SDD-AGE. Western blot detected all the truncated LEF-10-Sup35MC expressing at similar levels as full-length Sup35. PGK1 was measured as the loading control. Strains with Sup35 and Sup35MC were used as positive and negative controls. **c** The [LEF$^+$] phenotype of LEF-10$_{1-41}$ and LEF-10$_{12-34}$ is curable by deletion of the *HSP104* gene

(Supplementary Fig. 8). These results suggest that LEF-10 may be predisposed to switching to an aggregate state when an insect cell is infected by multiple virions, and the aggregation of LEF-10 will lead to the inactivation of the protein resulting in the reduction of the downstream gene expression that is dependent on it. Using the *p10* promoter-driven mCherry as a reporter for baculovirus late gene expression (Fig. 1a and Supplementary Fig. 1e), we found that for both the wild-type LEF-10 and the LEF-10$^{L21A}$ mutant, the expression levels of mCherry in cells with LEF$^{Ag}$ were very low, but in contrast, all the cells expressing mCherry at high levels displayed cells with LEF$^d$ phenotype (Fig. 6b). These results demonstrated that LEF-10 can exist in two different conformations in virus-infected insect cells, and high-level expression of the protein induced the switch to the aggregate state which resulted in the inhibition of late gene expression.

## Discussion

The data we present in this study shows that (a) LEF-10 can functionally replace the PrD of the yeast Sup35 protein allowing it to behave as a prion in yeast, and (b) that the LEF-10 protein can exist in two different conformations in insect cells. Such in vivo behavior is entirely consistent with LEF-10 being a baculovirus-encoded prion-forming protein. To date, all previously identified prions have been reported in cellular life forms. Whether viruses, the most abundant and diverse of life forms, might use prions was unknown and therefore our discovery extends prions to the viral world.

Studies in yeast have revealed that many prion proteins contain high aggregation tendency domains that are commonly enriched for glutamine and asparagine residues. Based on these findings, several PrD prediction algorithms have been established[34,36–39], which in turn have led to the discovery of more prion candidates

with Q/N-rich domains[7,11,12]. Contrary to this common characteristic, functional PrDs have also been described in non-Q/N-rich prion proteins e.g., HET-s[9], Mod5[8] and as shown in this report, LEF-10. These atypical PrDs do not share consensus primary sequences nor bias towards specific amino acid residues. As experimental identification of a prion protein is time consuming, especially on a genome-wide scale, development of a universal computational algorithm is a necessary task. This work is a challenge and partially depends on the discovery of more non-Q/N-rich prion proteins. Here, we provide another such instance that will inform future developments in these algorithms.

The majority of yeast prions are cytoplasmically-located and show non-Mendelian inheritance, however, virus-encoded prions are more likely to be associated with the regulation of virus replication due to the non-cellular nature of viruses. When baculoviruses infect insect cells at a high multiplicity and the expression level of LEF-10 reaches a threshold, the protein tends to form aggregates leading to a shutdown of the downstream gene expression. Such a regulatory mechanism may benefit the population transmission of baculovirus (Fig.6c). Surprisingly, being a sensor of this regulatory system, LEF-10 is more sensitive to form prion state than one might have expected. The blocking effect by LEF-10 only emerged at MOI of lower than 1 (Supplementary Fig. 8) and it is well established that a high MOI will reduce the production of recombinant protein using baculovirus as the expression platform[40] (although there is a contrary report for Orgyia pseudotsugata multicapsid nuclear polyhedrosis virus[41]). Based on our findings, we postulate that LEF-10 is a self-limiting factor that blocks virus late gene expression at high MOI, and release of LEF-10-derived restriction might be an efficient way to improve the productivity of baculovirus expression vector system.

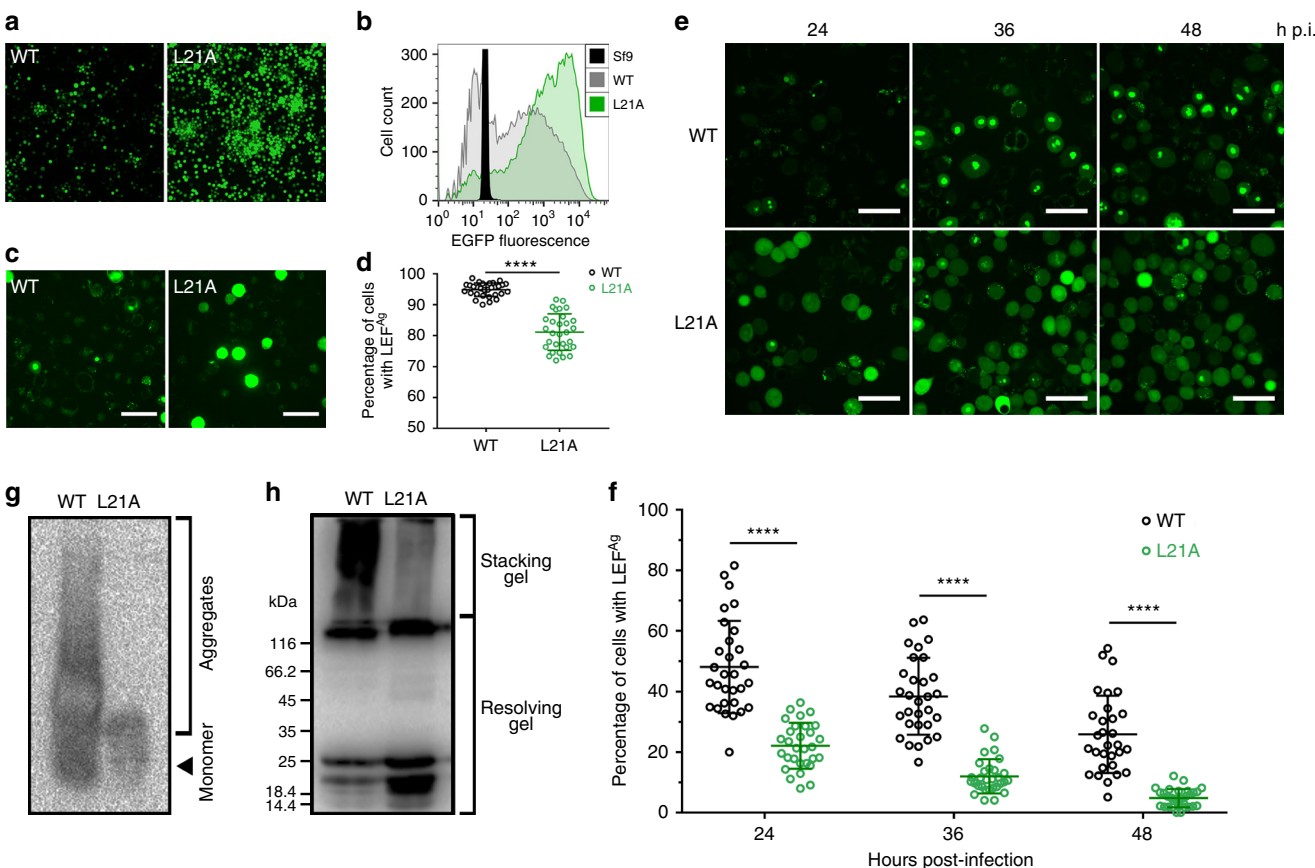

**Fig. 5** Characterization of the aggregates of LEF-10 and LEF-10[L21A] in virus-infected *Sf*9 cells. **a** Fluorescence microscope images displayed that wild-type LEF-10 and LEF-10[L21A] fused with EGFP were driven by a tandem *actin/p10* promoter (Supplementary Fig. 1b) and they could rescue the BacmidΔ*lef-10* (more mutants in Supplementary Fig. 4a). **b** Flow cytometry analysis revealed that the expression level of LEF-10[L21A]-EGFP was higher than that of LEF-10-EGFP. **c** Laser confocal microscopy images of over-expressed LEF-10-EGFP and LEF-10[L21A]-EGFP in infected *Sf*9 cells at 48 hpi (Related to Supplementary Fig. 4c). Both diffuse fluorescence and non-diffuse fluorescence could be observed in *Sf*9 cells expressing virus-encoded LEF-10-EGFP and LEF-10[L21A]-EGFP. Scale bar, 50 μm. **d** Percentage of cells which harbored puncta fluorescence formed by over-expressed LEF-10-EGFP or LEF-10[L21A]-EGFP were examined from 30 individual fields under laser confocal microscopy images. Bars represented mean ± SD. Two-tailed unpaired Student's *t*-test was performed (****$P \leq 0.0001$). Source data are provided as a Source Data file. **e** Laser confocal microscopy images of LEF-10-EGFP and LEF-10[L21A]-EGFP under the control of native *lef-10* promoter (Supplementary Fig. 1e) in infected *Sf*9 cells at 24, 36 and 48 hpi, MOI = 10. Scale bar, 50 μm. **f** Percentage of cells which harbored puncta fluorescence formed by LEF-10-EGFP or LEF-10[L21A]-EGFP under the control of native *lef-10* promoter (Supplementary Fig. 1e) were examined from 30 individual fields of laser confocal microscopy images at the indicated hours post-infection. Bars represented mean ± SD. Two-tailed unpaired Student's *t*-test was performed (****$P \leq 0.0001$). Source data are provided as a Source Data file. **g** SDD-AGE analysis of aggregates formed by LEF-10-EGFP-His and LEF-10[L21A]-EGFP-His fusion proteins in infected *Sf*9 cells at 48 hpi. Both chimeras were able to form such aggregates, albeit with different polymer size distribution. **h** Western blot analysis of aggregates formed by LEF-10-EGFP-His and LEF-10[L21A]-EGFP-His fusion proteins in infected *Sf*9 cells at 48 hpi. The high molecular-weight fraction remained in the stacking gel, and the low molecular-weight fraction and monomers were detected in the resolving gel. The LEF-10-EGFP-His and LEF-10[L21A]-EGFP-His proteins in **g** and **h** were detected using α-His antibody

The regulation of gene expression can occur at various levels. Altering protein activity through allosteric changes in a protein is no doubt a rapid way to regulate downstream gene expression, especially for viruses which have very short generation time. As parasitic organisms, the propagation of viruses depends on the environment provided by their hosts. The switchable conformation of a prion-forming protein could allow viruses to quickly respond to the stresses coming from their hosts. This advantage may encourage screening and retention of prion-like proteins during the rapid evolution of the virus. We further speculate that proteins with prion properties are unlikely to be rare in the virus world and more virus-encoded prion-forming protein would be discovered in the future.

Cross-seeding is a phenomenon whereby a heterologous aggregated protein can template the de novo conversion of another protein to switch to an aggregated form[29,42,43]. Several examples of human disease-related cross-seeding have been reported. For example, α-synuclein aggregates can promote the aggregation of Tau protein in neurons[42]. Amyloid beta (Aβ) can accelerate the aggregation of Tau in vivo[44] and it can also act as a seed to promote the aggregation of α-synuclein in vitro[45]. Furthermore, amyloid protein made by microbiota has been reported to enhance α-synuclein aggregation in aged rats and nematodes[46]. This accumulating evidence of widespread heterologous cross-seeding in amyloid formation raises the possibility that pathogen-derived aggregation-prone proteins may also trigger amyloid formation and hence disease in humans. Researchers have long noticed a correlation between viral infection and Alzheimer's disease[47], and recent reports further confirmed this connection[48,49]. Although it is too early to generalize, with our discovery of the first virus-derived prion, more attention should now be focused on whether virus-encoded prions can trigger the biogenesis of cellular prion-related diseases following virus infection in animals, especially in humans.

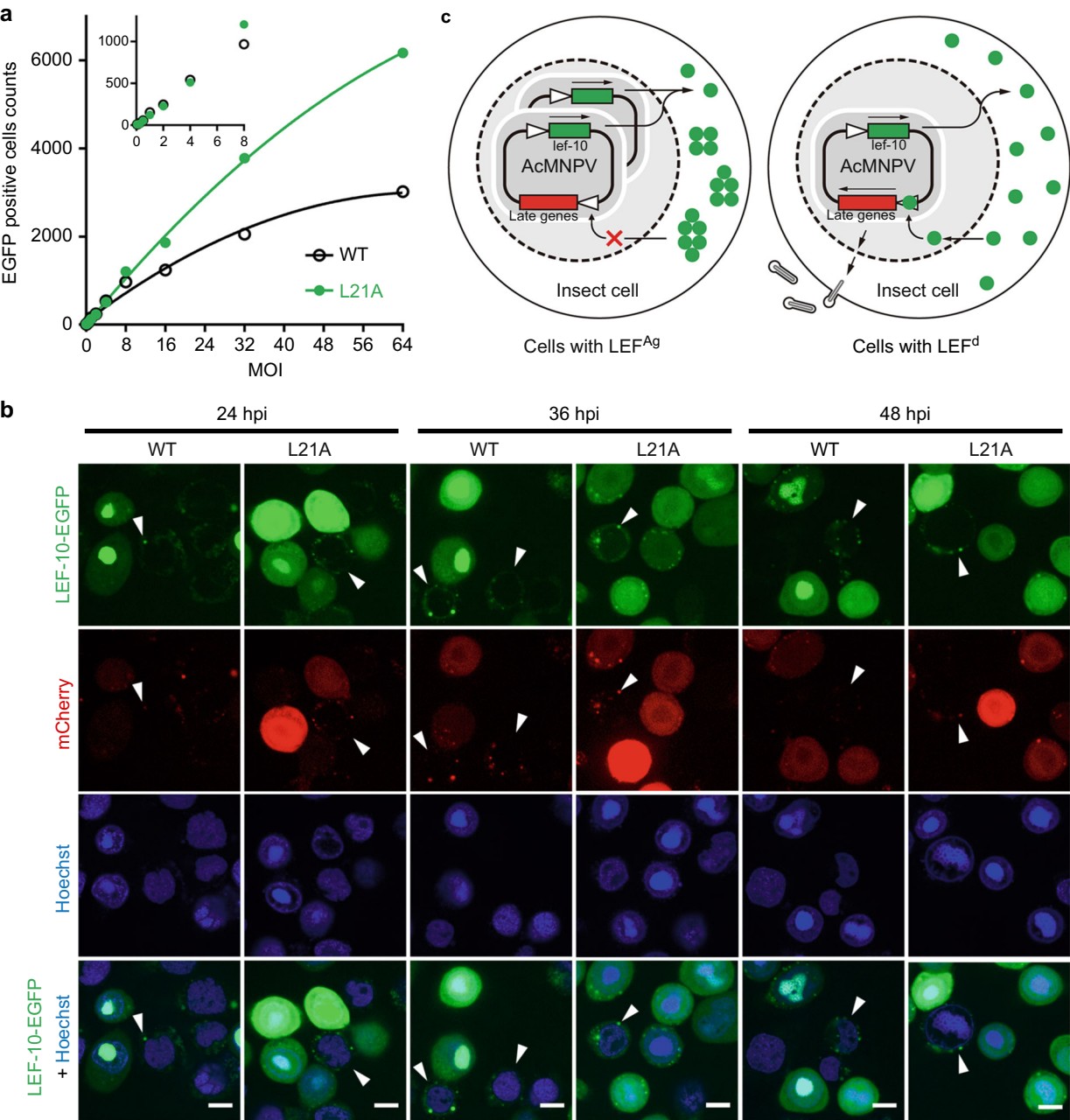

**Fig. 6** The viral late gene regulation mediated by LEF-10. **a** *Sf*9 cells were infected in parallel with serial dilutions of recombinant baculoviruses encoding wild-type LEF-10 or the LEF-10[L21A] mutant at the indicated MOI. EGFP served as the reporter of late gene expression (Supplementary Fig. 2d). Infected *Sf*9 cells expressing EGFP were gated and counted at 36 hpi. Source data are provided as a Source Data file. **b** Laser confocal microscopy images of infected (MOI = 10) *Sf*9 cells containing virus-encoded LEF-10 and LEF-10[L21A] at 24, 36 and 48 hpi. The recombinant baculoviruses produced LEF-10-EGFP or LEF-10[L21A]- EGFP under the control of native *lef-10* promoter and harbored a p10 promoter driven mCherry as a reporter for baculovirus late gene expression (Supplementary Fig. 1e). Hoechst stains the nuclei. Scale bar, 50 μm. **c** Model of how the LEF-10 endows prion-mediated regulation on late genes, resulting in the limitation of viral replication

## Methods

**Yeast strains and insect cells**. *Saccharomyces cerevisiae* strain LJ14[50] (*MATa ade1-14 trp1-289 his3D-200 ura3-52 leu2-3,112 SUP35::loxP* p[*SUP35-URA3*] [*PSI+*]) was used for shuffling experiments and phenotypic assays. LJ14-derived strains made in this study for yeast phenotypic experiments are listed in Supplementary Table 2. Standard rich (YPD) and Synthetic Dropout (SD) Media were used to culture yeast cells at 30 °C[51]. *Spodoptera frugiperda* (*Sf*9) cells (purchased commercially from the ATCC) were cultured in SFX insect medium (HyClone, GE Healthcare Life Sciences) supplemented with 1% fetal bovine serum (FBS) at 27 °C.

**Plasmids and Bacmid**. The viral *lef-10* gene sequence was amplified from Bacmid[52]. The plasmids derived from pTriEx1.1 vector (Merck), which carry *lef-10*

and its alleles and/or reporter gene expression cassettes and flanking sequences for the generation of recombinant baculoviruses by homologous recombination, are listed in Supplementary Table 1. pTriEx-EGFP was constructed by PCR amplification of the *egfp* gene with *Eco*RI and *Xho*I ends and was cloned into pTriEx1.1. The *lef-10* promoter and its coding sequence were cloned into this plasmid as a *Sac*II-*Xba*I fragment and the construct was named as pTriEx-LEF-10/EGFP. Site-directed mutagenesis of plasmids pTriEx-LEF-10-EGFP[25], pTriEx-P_*lef-10*-LEF-10-EGFP[25], pTriEx-LEF-10/EGFP, and pTriEx-P_*lef-10*-LEF-10-EGFP-P_*p10*-mCherry[25] were created with the Q5® Site-Directed Mutagenesis Kit (NEB).

To construct pUKC1620-derived plasmids (listed in Supplementary Table 1), varietal LEF-10 coding sequences were cloned into pUKC1620[50] as an *Xba*I-*Eco*RI fragment under the control of the *SUP35* promoter in yeast cells. The plasmid p6431(CUP1, GFP) was donated by Susan Lindquist. The *lef-10* and *lef-10(L21A)*

fragment were cloned into plasmid p6431 as a SacII-NheI fragment in-frame with the GFP under the control of the yeast CUP1 promoter. All successfully cloned plasmids (Supplementary Table 1) were confirmed by sequencing. Primers sequences are listed in Supplementary Table 4.

**Computational predication of protein aggregation propensity.** The aggregation propensity predication for late expression factor LEF-10 (NP_054083.1) of Autographa californica multiple nucleopolyhedrovirus (AcMNPV) was analyzed by two different programs: (1) PLAAC[34] (http://plaac.wi.mit.edu), a Hidden Markov Model (HMM)-based prion prediction algorithm for identifying probable prion subsequences in protein sequences, with setting minimal contiguous prion-like domain length of 10 and relative weighting of background probabilities ($\alpha$) of 50; and (2) TANGO[35] (http://tango.switchlab.org), a computer algorithm for the prediction of aggregating regions in unfolded polypeptide chains under standard conditions (pH = 7.0 and temperature [K] = 298.15).

**Plasmid shuffling and phenotypic analysis.** Plasmids containing LEF-10-Sup35MC variants were, respectively, transformed into strain LJ14 by the standard PEG/LiAc/ssDNA transformation method[53]. The resulting transformants were incubated on histidine-deficient and uracil-deficient synthetic medium for 3 days at 30 °C and then plated on YPD medium supplemented with 1 mg/ml of 5-fluoroorotic acid (CAS: 703-95-7, Sangon Biotech) to eliminate the Sup35 maintainer plasmid p[SUP35-URA3], generating strains expressing a LEF-10-Sup35MC fusion protein as the only source of functional Sup35. Phenotypes were assessed by growth of colonies overnight at 30 °C in YPD medium and streaking or spotting of different dilutions of these cultures on rich 1/4 YPD agar and rich 1/4 YPD agar with 5 mM guanidine hydrochloride[54].

Post-shuffled yeast strains were plated on SD-Ade medium to verify the read-through status of the premature UGA stop codon in the ade1-14 allele. In [PSI+] cells, the majority of Sup35 aggregated and was unavailable for translation termination, resulting in the read-through of ade1-14 premature stop codon. The synthesis of full length Ade1 protein (N-succinyl-5-aminoimidazole-4-carboxamide ribotide synthetase) resulted in white colonies on rich medium and growth on SD-Ade medium. In contrast, [psi-] cells showed no read-through of the ade1-14 premature stop codon and failed to grow on SD-Ade medium.

pUKC1620-LEF-10, a plasmid encoding a LEF-10-Sup35MC chimera substituting for full-length Sup35, was transformed in parallel with pUKC1620 containing the full-length SUP35 gene, into strain LJ14. Phenotypes of the post-shuffled colonies were assessed as described above. [PSI+]-like (i.e., white) and [psi-]-like (i.e., red) colonies were, respectively, passaged on YPD plates and re-streaked on SD-Ade medium to test the stability and maintainability of these two phenotypes.

**Generation of an HSP104 gene deletion.** A recombination-based PCR-generated deletion strategy[55] was used to delete the endogenous HSP104 gene in yeast. A replacement cassette containing the endogenous URA3 gene and its promoter fragment was amplified from p6431 using primers (listed in Supplementary Table 4), so that the replacement cassette was attached to two flanking homologous arms of the HSP104 sequence. The linear replacement fragment was transformed into recipient strains expressing either Sup35, LEF-10-Sup35MC, LEF-10$_{1-41}$-Sup35MC, or LEF-10$_{12-34}$-Sup35MC (listed in Supplementary Table 2) by standard PEG/LiAc/ssDNA transformation[53]. The resulting transformants were grown on SD-Ura medium for 3 days at 30 °C and then plated or streaked onto 1/4 YPD and/or SD-Ade media.

**Characterization of truncations and mutants of LEF-10.** Sup35MC-based assays described above were performed to define the candidate prion domain of LEF-10. To assess the prion characteristics, post-shuffled yeast strains carrying plasmid-encoded truncated LEF-10-Sup35MC, namely LEF-10$_{1-41}$-Sup35MC, LEF-10$_{35-62}$-Sup35MC, LEF-10$_{54-78}$-Sup35MC, and LEF-10$_{12-34}$-Sup35MC (listed in Supplementary Table 2), were plated or streaked on 1/4 YPD and SD-Ade media. Ten post-shuffled yeast strains containing LEF-10 mutants (L12A, I13A, V16A, I17A, N20A, L21A, L23A, I24A, I29A, and V33A) fused with Sup35MC were generated to assess the effects of conserved amino acid residue mutation on the prion properties.

**Generation of recombinant baculoviruses.** The recombinant baculoviruses in this research were generated by homologous recombination between BacmidΔlef-10[25] and pTriEx1.1-derived plasmids (Supplementary Fig. 1) in co-transfected Sf9 cells using FuGENE® HD Transfection Reagent (Promega). The baculoviruses supernatants were collected at 4–5 days post co-transfection and the supernatants were centrifuged at 500 × g for 5 min to remove cell debris. Then the baculoviruses titers were measured and infected Sf9 cells at specific MOI.

To investigate if the over-expressed mutant LEF-10 could rescue the lef-10 null bacmid, pTriEx-LEF-10-EGFP and 10 mutant constructs (Supplementary Table 1) were co-transfected with linearized lef-10 knockout AcMNPV bacmid into Sf9 cells. LEF-10-EGFP and its 10 mutants were driven by two tandem promoters consisting of a chicken actin promoter and the p10 promoter. EGFP fusion proteins reported whether the LEF-10 mutants could rescue the lef-10 knockout AcMNPV bacmid (Supplementary Fig. 4a and Fig. 5a, b). To investigate the aggregates formed by

LEF-10 and LEF-10$^{L21A}$ under the control of the native lef-10 promoter, pTriEx-LEF-10-EGFP-His and pTriEx-LEF-10$^{L21A}$-EGFP-His (Supplementary Table 1) were co-transfected with linearized lef-10 knockout AcMNPV bacmid into Sf9 cells. EGFP fusion proteins reflected the aggregates of LEF-10 or LEF-10$^{L21A}$. To visualize the expression of LEF-10 and investigate the effects of LEF-10 aggregation on viral late gene expression, two recombinant baculovirus vAc/P$_{lef-10}$-LEF-10-EGFP-P$_{p10}$-mCherry and vAc/P$_{lef-10}$-LEF-10$^{L21A}$-EGFP-P$_{p10}$-mCherry (Fig. 1 and Fig. 6b) were generated by homologous recombination between the linearized lef-10 knockout AcMNPV bacmid and the plasmids pTirEx-LEF-10-EGFP/mCherry or pTirEx-LEF-10$^{L21A}$-EGFP/mCherry (Supplementary Table 1). EGFP fusion proteins under the control of the native lef-10 promoter displayed the aggregate status of LEF-10 or LEF-10$^{L21A}$, and mCherry reported the expression level of viral late genes. In order to further investigate the influence of LEF-10 aggregation on late gene expression, two recombinant baculovirus vAc/P$_{lef-10}$-LEF-10-His-P$_{p10}$-EGFP and vAc/P$_{lef-10}$-LEF-10$^{L21A}$-His-P$_{p10}$-EGFP (Supplementary Fig. 1c, Fig. 6a, Supplementary Fig. 7 and Supplementary Fig. 8) were generated by homologous recombination between linearized lef-10 knockout AcMNPV bacmid and the plasmids pTirEx-LEF-10/EGFP or pTirEx-LEF-10$^{L21A}$/EGFP (Supplementary Table 1). EGFP under the control of the p10 promoter reported the late gene expression.

**One-step growth curve.** Sf9 cells were infected with recombinant baculoviruses vAc/P$_{lef-10}$-LEF-10-P$_{p10}$-EGFP and vAc/P$_{lef-10}$-LEF-10$^{L21A}$-P$_{p10}$-EGFP (Supplementary Table 3) at an MOI of 0.5. The baculoviruses supernatants of infected Sf9 cells were harvested at 12, 24, 48, 72, 96, 120, and 144 hpi (hours post-infection), respectively, and the supernatants were centrifuged at 500 × g for 5 min to remove cell debris. Based on the 50% tissue culture infective dose (TCID$_{50}$) endpoint dilution assay, the Reed-Muench method[56] was used to quantify virus titers. Each one-step growth curve experiment was replicated three times.

**Flow cytometry.** Sf9 cells were harvested at 36 hpi (Fig. 6a and Supplementary Fig. 7b), 48 hpi (Fig. 5b and Supplementary Fig. 4a) or 60 hpi (Supplementary Fig. 8b, c) by centrifugation at 300 × g for 2 min, and the pellet was resuspended in PBS. Data detection and collection were performed on a CyFlow Cube 6 flow cytometer (PARTEC) at an excitation wavelength of 488 nm. Cells expressing EGFP fusion proteins were detected in the FL1 channel at an emission wavelength of 530 nm. Data were collected from at least 10,000 cells for each sample and cell populations were analyzed offline using FlowJo® software. The infected Sf9 cells expressing EGFP (EGFP positive cells) were gated (Supplementary Fig. 7b) and EGFP positive cells were counted (Fig. 6a) at 36 hpi. Infected Sf9 cells were gated (Supplementary Fig. 8b) and the mean fluorescence intensity (MFI) was calculated (Supplementary Fig. 8c) at 60 hpi. Second Order Polynomial regression was performed for regression analysis using GraphPad Prism version 7.00 for Windows, GraphPad Software, La Jolla California USA, www.graphpad.com.

**Cell imaging.** The plasmid p6431-LEF-10 and p6431-LEF-10$^{L21A}$ (Supplementary Table 1) were individually transformed into the yeast strain LJ14-SUP1 (Supplementary Table 2) using the standard PEG/LiAc/ssDNA transformation method[53]. When yeast cells cultured in SD-Ura medium reach logarithmic growth phase, CuSO$_4$ was added to a final concentration of 25 μM to induce the expression of GFP fusion proteins. Yeast cells were imaged at 4 h post-induction with a Revolution XD spinning disk confocal system equipped with a CSU-W1 spinning-disk head (Yokogawa) and an iXon Ultra 888 EMCCD (Andor) on a DMi8 microscope body (Leica). Specifically, yeast cells were imaged with a HCX PL Apo 100×/1.44 oil immersion objective (Leica). GFP was excited at 488 nm and a TR-F525/50 bandpass emission filter (Semrock Brightline) were used for capturing GFP signals. Images were captured with Andor iQ3 software (Andor). Confocal images were cropped and processed with the ImageJ Software.

Sf9 cells were infected with the recombinant baculovirus vAc/P$_{lef-10}$-LEF-10-EGFP-P$_{p10}$-mCherry (Supplementary Table 3) at an MOI of 3 (Fig. 1). After incubation at 27 °C for 60 h, the expression of LEF-10-EGFP and mCherry were detected by DM5000 B fluorescence microscope (Leica) at an emission wavelength of 530 nm or 585 nm (Fig. 1b). Images were captured with Leica Application Suite software (Leica).

Sf9 cells were infected with recombinant baculovirus vAc/P$_{actin-p10}$-LEF-10-EGFP and its 10 variants (Supplementary Table 3) were observed at 48 hpi with Revolve R4 fluorescence microscopy (Echo Laboratories) at an emission wavelength of 530 nm (Supplementary Fig. 4a). Images were captured with Echo app (Echo Laboratories).

Sf9 cells were infected with recombinant baculovirus vAc/P$_{lef-10}$-LEF-10-P$_{p10}$-EGFP and vAc/P$_{lef-10}$-LEF-10$^{L21A}$-P$_{p10}$-EGFP at a serial 2-fold dilutions of MOI as 0.125 to 4 (Supplementary Fig. 7) and 0.0625- 64 (Supplementary Fig. 8). Infected Sf9 cells were observed at 36 hpi (Supplementary Fig. 7) with DMi3000 B inverted fluorescence microscope (Leica) equipped with a DFC365 FX CCD (Leica). Infected Sf9 cells were observed at 60 hpi (Supplementary Fig. 8) with DMi8 inverted fluorescence microscope (Leica) equipped with DFC9000 GT sCMOS camera (Leica) at an emission wavelength of 530 nm. Images were captured with Leica Application Suite X software (Leica).

Sf9 cells were infected with recombinant baculovirus vAc/P$_{actin-p10}$-LEF-10-EGFP and vAc/P$_{actin-p10}$-LEF-10$^{L21A}$-EGFP (Supplementary Table 3) at an MOI of

10. After incubation at 27 °C for 48 h, cell images were captured with Revolution XD spinning disk confocal system equipped with a CSU-W1 spinning-disk head (Yokogawa) and an iXon Ultra 888 EMCCD (Andor) on a DMi8 microscope body (Leica). *Sf*9 cells were infected with recombinant baculovirus vAc/$P_{lef-10}$-LEF-10-EGFP-$P_{p10}$-mCherry and vAc/$P_{lef-10}$-LEF-10$^{L21A}$-EGFP-$P_{p10}$-mCherry (Supplementary Table 3) at an MOI of 10. Cells were imaged at 12, 36, and 48 hpi with the spinning disk confocal system described above. Specifically, infected *Sf*9 cells were imaged with a HCX PL Apo 63 × /1.32 oil immersion objective. EGFP was excited at 488 nm. mCherry was excited at 561 nm. Hoechst 33342 Fluorescent Stain was excited at 405 nm. A TR-F525/50, a TR-F593/46 and TR-F447/60 bandpass emission filter (Semrock Brightline) were used for capturing EGFP, mCherry and Hoechst stain signals, respectively. Images were captured with Andor iQ3 software (Andor). Confocal images were cropped and processed with the ImageJ Software.

**Immunoblotting**. To investigate the amyloid propensities of LEF-10, LJ14-derived yeast strains harboring various LEF-10-Sup35MC chimeras under the control of the endogenous *SUP35* promoter were cultured to mid-exponential phase and then processed for semi-denaturing detergent-agarose gel electrophoresis (SDD-AGE) analysis[30].

The expression of truncated LEF-10-Sup35MC fusion proteins was detected by Western blot, probing with a Sup35C-specific antibody (1:500, #sc-25915, Santa Cruz Biotechnology). Endogenous phosphoglycerate kinase 1 (PGK1) was detected with a PGK1-specific antibody (1:2000, #GTX107614, GeneTex) and served as a loading control.

In virus rescue assays, LEF-10-EGFP variants in the lysates of baculovirus-infected *Sf*9 cells were detected by Western blot (Supplementary Fig. 4b) using a GFP-specific antibody (1:1000 #AF0159, Beyotime). Infected *Sf*9 cells harboring baculovirus-encoded LEF-10-EGFP-His and LEF-10$^{L21A}$-EGFP-His under the control of native *lef-10* promoter were analyzed by Western blot (Fig. 5h) and SDD-AGE (Fig. 5g) using a His-tag specific antibody (1:5000, #CW0286M, CoWin Biosciences) at 48 hpi. Endogenous β-tubulin was detected with a β-tubulin specific antibody (1:2000, #CW0098M, CoWin Biosciences) and served as a loading control. HRP-conjugated goat anti-rabbit IgG (1:5000, CW0103S, CoWin Biosciences), HRP-conjugated rabbit anti-goat IgG (1:5000, CW0105S, CoWin Biosciences) or HRP-conjugated goat anti-mouse IgG (1:5000, CW0102S, CoWin Biosciences) antibodies were used as secondary antibodies. Proteins were detected with Immobilon$^{TM}$ Western Chemiluminescent HRP Substrate (Millipore) and a ChemiDocXRS + imaging system (Bio-Rad). All uncropped scans of blots are provided as a supplementary figure in the Supplementary Information (Supplementary Fig. 9).

**Reporting summary**. Further information on experimental design is available in the Nature Research Reporting Summary linked to this article.

## Data availability

All relevant data are available from authors upon request. The source data underlying Figs. 5d, 5f, 6a and Supplementary Figs. 5 and 8c are provided as a Source Data file.

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

## Acknowledgements

We sincerely thank the late Professor Susan L. Lindquist (Whitehead Institute, Massachusetts Institute of Technology, USA) for providing plasmid p6431. We also sincerely thank Yujie Liu (NWAFU, China) for the excellent technical assistance with prion identification of LEF-10, and Xinlong Qiao (NWAFU, China) for the generous and fruitful help with yeast experiments. We are grateful to Professor Fei Yu (NWAFU, China) and Associate Professor Xiayan Liu (NWAFU, China) for providing effective coordination and valuable reagents, and Min Jia (NWAFU, China) for providing professional technical assistance with laser confocal microscopy imaging. We sincerely thank Professor Ian M. Jones (University of Reading, UK) for critical reading and modification on the manuscript. We thank the Teaching and Research Core Facility at College of Life Science, NWAFU for their support in this work. We thank Bacmid Ltd. (Shaanxi, China) for providing reagents and consumables for cells culture. This work was supported by NWAFU Start-up grant and prion research in the M.F.T. laboratory was supported by the Biotechnology and Biological Sciences Research Council (BBSRC) grant ref nos: BB/H012982/1 and BB/J000191/1.

## Author contributions

X.X. conceived the project; X.X. and H.N. designed the research; X.X. and H.C. supervised the experiments; H.N. performed the experiments; M.F.T. provided technical guidance in yeast studies; X.X., H.C., and H.N. analyzed the data and all authors discussed the results and co-wrote the manuscript.
