## [Peer Review File · Nature Communications]

Reviewers' Comments:

Reviewer #1:

Remarks to the Author:

This is an interesting article describing a viral protein LEF10 that can undergo a conformational change from a soluble and functional form to an aggregated and non-functional form. Using the classic Sup35MC reporter assay the authors showed that the Sup35MC-fusion of this viral protein can exhibit many [PSI⁺]-like phenotypes in budding yeast. More interestingly, the authors showed that a small region containing only 23 amino acid residues can fully exhibit all prion-like features and the [PSI⁺]-like phenotypes are dependent on the Hsp104 function. Until to this point, the described work is beautifully done, very solid. However, the argument that the L21 residue is responsible for prion-mediated aggregation is not convincing. In Fig 6a, cells carrying L21A mutant of Sup35MC fusion do not look redder in my eyes. The fact that [PSI⁺]-like cells of LEF-10L21A-Sup35MC exhibited similar Ade⁺ growth phenotype casts another doubt on the claim that L21A is important for prion-mediated aggregation. Data gathered from Fig 6 and Fig 7 seem to suggest that L21A is more functional than the WT in terms of viral infection. However, more evidence is needed to tie its effect to prion aggregation. Although Fig 6d seems to suggest that LEF-10L21A-Sup35MC is monomer whereas the WT form is aggregated in viral infected cells, this image has a rather poor resolution. It does not make sense that the α -His gives a strong band but the smeared monomer band has a weak signal? Other concerns:

- 1) Abstract – line 11: change "..., a well-known yeast prion," to "..., a well-known yeast prion protein,"
- 2) Line 19 – change ", misfolded prion proteins..." to ", misfolded prion proteins (PrP)..."
- 3) Line 22 – change "... [PIN+]..." to "... [PIN+] or also known as [RNQ+]...". Also cite the article by Sondheimer and Lindquist, 2000, Mol. Cell.
- 4) Line 23 – remove [ISP+] and its reference as it is confirmed now [ISP+] is not a prion rather an aneuploidy artifact.
- 5) Line 23 – remove [C] and its reference.
- 6) Line 33 – the article by Cai et al 2014 Cell should be cited when discussing functional amyloids/prions.
- 7) Line 90 – reference 28 was wrongly cited and should be removed.
- 8) What is the [PIN+]/[RNQ+] prion background of the yeast strain used for Sup35MC assay? Is the [PSI⁺]-like conversion dependent on the [PIN+]/[RNQ+] prion background? This should be discussed.
- 9) Line 143, reference 3 is wrongly cited and should be removed.
- 10) Line 150 – reference 30 and 33 are wrongly cited and should be removed. Please cite the review by Liebman and Chernoff, 2012 and the review by Crow and Li 2011.
- 11) Fig 5b, why there are two bands in the Western blot when probed with α -Sup35C?
- 12) Fig 5 legend, remove the last part since it was wrongly copied and pasted from Fig 4 legend.
- 13) line 343 – reference 46 or 47 does not belong there.

Reviewer #2:

Remarks to the Author:

Summary

Previous studies by this group has shown that overexpression of LEF10 can result in aggregation and suppression of baculovirus late gene expression. In addition, expression under the native lef-10 promoter can result in a subpopulation of cells where LEF10 will aggregate and appears to be correlated with a decrease in late gene expression. In this study the authors utilize a yeast prion assay

to suggest that LEF10 functions as a prion to regulate gene expression. A LEF10-Sup35 fusion, with LEF10 replacing the known Sup35 prion domain was constructed. The assay was able to show that LEF10 could functionally replace the Sup35 domain for duplicating the [PSI+] phenotype and that it was dependant on HSP104. Similarly, the yeast assay was used to map the LEF10 domain to amino acids 14 to 34. Point mutations in conserved sites failed to identify essential amino acids in the yeast assay. It was suggested that L21 may diminish the prion phenotype. Recombinant AcMNPV viruses were constructed in a *lef10*-null backbone that expressed LEF mutants and L17, N20 and I29 failed to rescue virus replication. The LEF10 L21 mutant was observed not to form aggregates. Differential MOI infections with the L21 mutant compared to WT suggested that that late gene expression was not down regulated.

Comments

The results of this are a reasonable extension of the PLoS One 11,e0154835 study by these authors however there is some overlap between these studies. Overall the data suggests that in the yeast assay at least, LEF10 appears to functionally replace the Sup35 prion domain. However this phenotype beyond aggregation has not been shown in viral infected insect cells or in insect cells in the absence of infection.

Specific comments

Line 43-45. Convolutd sentence, rewrite.

Figure 1, 6, and 7? The Materials and Methods describing how the viruses were constructed needs to be clearly reported in detail. The source and origin of all viral DNAs and details of the repair vectors and how they were constructed need to be shown or references that give precise details. It would appear, but it's extremely hard to tell, that none of the viruses are clones. If the viruses were made by co-transfection with linear viral DNA, the use of supernatants is not appropriate. Viruses need to plaque purified.

Line 45-48. AcMNPV is the archetype species of the genus alphabaculovirus of the family Baculoviridae. Species name "*Autographa californica nucleopolyhedrovirus*" should all be written in italics.

Figure 1. What was the MOI, how late in infection is this image recorded? What percentage of cells has this phenotype? Does the proportion of cells showing aggregate formation increase with time? The authors need to define "percentage of pixel". How is it known the foci of LEF10-GFP are insoluble?

Line 92. Define the [PSI+] and [psi-] phenotypes.

Line 169-172. Alignment does not confirm a "discrete cPrD", it only identifies potentially conserved (not "constant") domains. Alignment is on 28 baculoviruses not "nucleopolyhedroviruses". Alignment was previously reported by the authors and it could be referenced instead of shown and only the primary sequence of LEF10 provided.

Line 190. Delete "Without exception", there are only 5 constructs and 3 have the domain.

Line 211. "Interestingly, the LEF-10 cPrD aligned with the most conserved region of C1" This sentence does not make sense, since the alignment is of homologs of AcMNPV LEF10?

Line 211-215. Awkward run-on sentence rewrite.

Line 215-218. The figure 6a does not show any quantitative reduction in efficiency of ade1-14.

Lines 219-226. The authors need to make it clear they are now discussing recombinant viruses and bacmid transfected insect cells. The actin/p10 is an artificial construct which the authors are trying to use as a marker for late gene expression. Actin is a constitutive promoter that expresses genes a relatively high level. It is unknown how when combined with p10, if the actin promoter affects p10 late gene regulation. Why wasn't the native p10 promoter used? It is claimed that there was significantly higher expression in L21 mutant. How was this determined and what was the quantitative assay that was used? There are no units or axis labels on the flow cytometry graphs. If there is an increase expression in the L21 mutant it could be due to increased DNA replication and copy number. This needs to be checked. In Fig. 6c there does not appear to be any significant difference, how many hours post-transfection were these samples analysed and how many repeats were performed and was the efficiency of transfection always similar and how was this confirmed? Any differences in expression could just reflect a productive infection and a functional LEF10. In Fig. 6d how many cells were analysed and at what times?

Line 227-229. The data as shown does not support the statement that, "the phenotype of LEF-10L21A-Sup35MC displayed redder color in 1/4 YPD medium than wild-type Sup35".

Line 260. What is meant by "innate weak promoter". How is the promoter defined as weak?

Figure 7. Statistically this experiment does not make sense. To ensure every cell gets infected an MOI of 5-10 must be used. In addition, it is assumed that significantly higher LEF10 expression will be achieved by having a cell infected with more genomes. Is there any evidence for this to be true? The assay is a late gene promoter which is activated after DNA replication is initiated and multiple genomes are present. Again, no time post-infection is provided. The authors need to perform higher MOI infections, ensuring every cell is infected and analyze and compare multiple time points.

Line 298-299. Data does not show LEF10 exists in alternative conformations in insect cells.

Line 316-317.." once viruses infect a cell at high multiplicity, the virus could limit its early replication to avoid the excessive stimulation of host antiviral responses"

What is meant by "early replication"? What evidence is there that baculoviruses cause excessive stimulation of an antiviral response? Most antiviral responses occur rapidly upon sensing a pathogen yet the events described in this paper occur late.

Line 319-321. This is not necessarily true. It depends on the protein and the cell type. Authors should also read, *Journal of General Virology* (1990), 71, 2841-284.

Line 322. The authors never used a high MOI.

Line 325. "rapid"

Reviewer #3:

Remarks to the Author:

This is an important paper with compelling evidence that the virus uses a prion protein for normal functional regulation. I suggest that the introduction should be more balanced about disease vs

benefit nature of prions. While generally well written the manuscript contains a surprisingly high number of typos and grammatical errors. The native English speaking author should look carefully for these errors.

The evidence that the viral domain acts as a prion in yeast is good. One addition that would be nice is to show either non-Mendelian segregation of cytoduction of this chimeric prion in yeast. I also had trouble understanding how the virus rescue assay relates to Fig. 6B and don't see the important redder color talked about for the 21A mutant. In that regard it would be nice (but not necessary) to see in yeast that the 21A mutation prevents formation of SDS resistant oligomers.

We would like to thank all three reviewers for the time they have taken to review our manuscript and for the extremely helpful and perceptive comments and corrections they make. Below we provide point-by-point responses to all of the comments made by the three referees and believe these changes – including new data added to the revised manuscript - significantly strengthen the conclusions we draw from our results.

Point-by-point responses to the referees' comments

Reviewer #1 (Remarks to the Author):

This is an interesting article describing a viral protein LEF10 that can undergo a conformational change from a soluble and functional form to an aggregated and non-functional form. Using the classic Sup35MC reporter assay the authors showed that the Sup35MC-fusion of this viral protein can exhibit many $[PSI^+]$ -like phenotypes in budding yeast. More interestingly, the authors showed that a small region containing only 23 amino acid residues can fully exhibit all prion-like features and the $[PSI^+]$ -like phenotypes are dependent on the Hsp104 function. Until to this point, the described work is beautifully done, very solid.

However, the argument that the L21 residue is responsible for prion-mediated aggregation is not convincing. In Fig 6a, cells carrying L21A mutant of Sup35MC fusion do not look redder in my eyes. The fact that $[PSI^+]$ -like cells of LEF-10^{L21A}-Sup35MC exhibited similar Ade⁺ growth phenotype casts another doubt on the claim that L21A is important for prion-mediated aggregation.

We have double checked the results in yeast and found that L21A mutant on YPD media displayed a slightly redder color than wild-type but the difference is not obvious (see image opposite). We agree that the L21A mutation did not abrogate the prion-mediated aggregation in yeast cells as $[PSI^+]$ -like cells of LEF-10^{L21A}-Sup35MC exhibited the Ade⁺ growth phenotype.

Therefore, to further investigate if L21 is important for prion-mediated aggregation in yeast cells or not, we expressed the GFP-tagged LEF-10 and LEF-10^{L21A} in yeast cells and found that LEF-10 and LEF-10^{L21A} exhibited different non-diffuse fluorescence patterns (revised Supplementary Figure 6a). In addition, we demonstrated by SDD-AGE that both GFP-tagged LEF-10 and LEF-10^{L21A} formed similar aggregates *in vitro* (revised Supplementary Figure 6b).

However, the conclusion that L21 is important for LEF-10 aggregation has also been made from new experiments using insect cells. In infected *Sf9* cells, we observed that over-expressed LEF-10^{L21A}-EGFP formed fewer aggregates than overexpressed LEF-10-EGFP (see revised Fig. 5c). To avoid the influence of LEF-10 overexpression on virus production, we further constructed a cassette that puts LEF-10 and LEF-10^{L21A} under the control of the native *lef-10* promoter (a weak promoter). By doing so we found that infected *Sf9* cells expressing LEF-10^{L21A}-EGFP exhibited a weak $[LEF^+]$ phenotype at 24 hpi (hours post-infection), 36 hpi

and 48 hpi (see revised Fig. 5e and 5f). Moreover, SDD-AGE analysis and Western blot revealed that L21A could form lower molecular-weight SDS-resistant polymers that migrated slower than the monomeric form (revised Fig. 5g and 5h).

The results allow us to conclude that L21 is associated with aggregation in infected *Sf9* cells and that the L21A mutant showed a reduced LEF-10 aggregation tendency. However, it seems that L21 is not responsible for prion-mediated aggregation in yeast cells. Since our description may cause misunderstanding, we have modified the relevant text in the revised manuscript.

Data gathered from Fig 6 and Fig 7 seem to suggest that L21A is more functional than the WT in terms of viral infection. However, more evidence is needed to tie its effect to prion aggregation.

We thank the reviewer for bringing this out. To clarify this point, we have added new data in the revised manuscript. Because LEF-10's function remains unknown, we employed *p10* promoter-driven EGFP to report the functional changes in the L21A mutant. To avoid the complication of acquiring the protein over-expression induced prion state, insect cells were infected at a multiplicity of infection (MOI) of 0.125. Infected cells were collected at 36 hpi and analyzed by flow cytometry. One hundred infected cells of each sample were gated and their EGFP fluorescence intensity was measured. Reporter gene expression levels in infected cells did not show significant difference ($P=0.2094$) (revised Supplementary Figure 7c). In addition, we have now measured the one-step growth curve of L21A mutant virus. In comparison with the wild-type virus, the L21A mutation in LEF-10 reduced the virus proliferation (see revised Supplementary Figure 5). This result implies that L21A is a virustatic mutant in infected insect cells. Together, our results suggest that L21A mutation does not upregulate LEF-10's physiological function in terms of viral infection.

Although Fig 6d seems to suggest that LEF-10^{L21A}-Sup35MC is monomer whereas the WT form is aggregated in viral infected cells, this image has a rather poor resolution.

We thank the reviewer for pointing this out. We have now imaged infected insect cells using a laser confocal microscope. Owing to the higher resolution and higher sensitivity of confocal microscopy, we have found that L21A did not abolish the aggregation property completely and some infected cells were found to harbor aggregates formed by LEF-10^{L21A} (see revised Supplementary Figure 4c). At high brightness/high contrast, the fluorescence signals of aggregates were captured and those cells harboring aggregates were counted. Doing so allowed us to demonstrate that overexpressed LEF-10^{L21A}-EGFP formed fewer aggregates than overexpressed LEF-10-EGFP in baculovirus-infected insect cells (see revised Figure 5c and 5d). Therefore, we replaced the original images with new images (see revised Fig. 5c and Supplementary Figure 4c) captured by confocal microscope and amended the relevant text and Figure Legend accordingly.

It does not make sense that the α -His gives a strong band but the smeared monomer band has a weak

signal?

We thank the reviewer for pointing this out. As agarose gels are used in SDD-AGE, the monomer band readily diffuses and we have noticed that the smeared monomer always gives weak signal. Furthermore, we have also observed that in Western blot analysis using SDS-PAGE, a portion of the protein (be it LEF-10-EGFP-His or LEF-10^{L21A}-EGFP-His) is retained in the stacking gel (see revised Fig. 5h) and more LEF-10-EGFP-His than LEF-10^{L21A}-EGFP-His is retained in the stacking gel. To avoid confusion, we have replaced the original Western blot image with a new one (see revised Fig. 5h) containing the protein bands in both the stacking gel and the resolving gel.

Other concerns:

1) Abstract – line 11: change “..., a well-known yeast prion,” to “..., a well-known yeast prion protein,”

Thanks for the suggestion. We have modified the text accordingly.

2) Line 19 – change “, misfolded prion proteins...” to “, misfolded prion proteins (PrP)...”

Thanks for your suggestion. We have modified the text accordingly.

3) Line 22 – change “... [*PIN*⁺]...” to “... [*PIN*⁺] or also known as [*RNQ*⁺]...”. Also cite the article by Sondheimer and Lindquist, 2000, Mol. Cell.

Thanks for your suggestion. We have modified the text accordingly and cited this article in the revised manuscript.

4) Line 23 – remove [*ISP*⁺] and its reference as it is confirmed now [*ISP*⁺] is not a prion rather an aneuploidy artifact.

We thank the reviewer for clarifying and pointing out this error, and we have removed [*ISP*⁺] and its reference from our revised manuscript.

5) Line 23 – remove [C] and its reference.

We thank the reviewer for pointing out this error and we have removed [C] and its reference from our revised manuscript.

6) Line 33 – the article by Cai et al 2014 Cell should be cited when discussing functional amyloids/prions.

Thanks for the suggestion. The reference has been cited in our revised manuscript.

7) Line 90 – reference 28 was wrongly cited and should be removed.

We thank the reviewer for pointing out this error and we have removed the reference from our revised manuscript.

8) What is the $[PIN^+]/[RNQ^+]$ prion background of the yeast strain used for Sup35MC assay? Is the [PSI⁺]-like conversion dependent on the $[PIN^+]/[RNQ^+]$ prion background? This should be discussed.

All yeast strains used in these experiments are $[PIN^+]/[RNQ^+]$. We don't know if the [PSI⁺]-like conversion is dependent on the $[PIN^+]/[RNQ^+]$ or not for now. Whether [PSI⁺]-like conversion is $[PIN^+]/[RNQ^+]$ -dependent is an issue focused on the LEF-10's behavior in a heterologous environment and what matters is how it behaves in the native host where there are no known sequence or functional orthologues of Rnq1.

9) Line 143, reference 3 is wrongly cited and should be removed.

We thank the reviewer for pointing out this error and we have removed the reference from our revised manuscript.

10) Line 150 – reference 30 and 33 are wrongly cited and should be removed. Please cite the review by Liebman and Chernoff, 2012 and the review by Crow and Li 2011.

We thank the reviewer for pointing out these errors and we have made the changes accordingly in the revised manuscript.

11) Fig 5b, why there are two bands in the Western blot when probed with α -Sup35C?

We speculate that Sup35MC and its fusion proteins may breakdown due to storage in freezer overnight. The bigger bands are full-length fusion proteins and smaller bands are breakdown proteins containing Sup35MC. When SDS-PAGE is immediately performed after sample preparation, only one band can be detected.

12) Fig 5 legend, remove the last part since it was wrongly copied and pasted from Fig 4 legend.

We thank the reviewer for pointing out this error and we have made the change accordingly.

13) line 343 – reference 46 or 47 does not belong there.

We apologize for the mistakes, and we have now removed the references from our revised manuscript.

Reviewer #2 (Remarks to the Author):

Summary

Previous studies by this group has shown that overexpression of LEF10 can result in aggregation and suppression of baculovirus late gene expression. In addition, expression under the native lef-10 promoter can result in a subpopulation of cells where LEF10 will aggregate and appears to be correlated with a decrease in late gene expression. In this study the authors utilize a yeast prion assay to suggest that LEF10 functions as a prion to regulate gene expression. A LEF10-Sup35 fusion, with LEF10 replacing the known Sup35 prion domain was constructed. The assay was able to show that LEF10 could functionally replace the Sup35 domain for duplicating the [PSI⁺] phenotype and that it was dependant on HSP104. Similarly, the yeast assay was used to map the LEF10 domain to amino acids 14 to 34. Point mutations in conserved sites failed to identify essential amino acids in the yeast assay. It was suggested that L21 may diminish the prion phenotype. Recombinant AcMNPV viruses were constructed in a lef10-null backbone that expressed LEF mutants and L17, N20 and I29 failed to rescue virus replication. The LEF10 L21 mutant was observed not to form aggregates. Differential MOI infections with the L21 mutant compared to WT suggested that that late gene expression was not down regulated.

Comments

The results of this are a reasonable extension of the PLoS One 11,e0154835 study by these authors however there is some overlap between these studies. Overall the data suggests that in the yeast assay at least, LEF10 appears to functionally replace the Sup35 prion domain. However this phenotype beyond aggregation has not been shown in viral infected insect cells or in insect cells in the absence of infection.

Thanks for the comments. We have conducted more experiments in viral infected insect cells as suggested by the reviewer. We hope that these new data have provided additional support to our main conclusions.

Specific comments

Line 43-45. Convolutud sentence, rewrite.

We thank the reviewer for pointing this out. We have modified the sentences in the revised manuscript.

Figure 1, 6, and 7? The Materials and Methods describing how the viruses were constructed needs to be clearly reported in detail. The source and origin of all viral DNAs and details of the repair vectors and how they were constructed need to be shown or references that give precise details. It would appear, but it's extremely hard to tell, that none of the viruses are clones. If the viruses were made by co-transfection with linear viral DNA, the use of supernatants is not appropriate. Viruses need to be plaque purified.

We thank the reviewer for the comments and suggestions. We have modified the Materials and Methods and added a diagram in Supplementary Figure 1 to give the details of the repair vectors and how they were constructed.

The bacmid used here lacks part of the essential gene ORF1629. The essential gene deletion prevents virus replication in insect cells and homologous recombination within the insect cells restores ORF1629 allowing the recombinant virus to replicate, so only recombinant viruses can grow and the viruses don't need to be plaque purified. (Reference: Zhao Y, Chapman DA, Jones IM. Improving baculovirus recombination. *Nucleic Acids Res.* 2003 Jan 15; 31(2): E6-6.)

Line 45-48. AcMNPV is the archetype species of the genus alphabaculovirus of the family Baculoviridae. Species name "*Autographa californica nucleopolyhedrovirus*" should all be written in italics.

Thanks for pointing this out. We have corrected this in the revised manuscript.

Figure 1. What was the MOI, how late in infection is this image recorded? What percentage of cells has this phenotype? Does the proportion of cells showing aggregate formation increase with time? The authors need to define "percentage of pixel". How is it known the foci of LEF10-GFP are insoluble?

In Fig.1, the *Sf9* cells were infected at a MOI of 3 and were imaged at 60 hpi. About one third infected cells showed aggregate phenotype. However, we had not provided accurate calculations. To address the concern about the proportion of cells showing aggregate formation, we observed the infected cells at 24, 36 and 48 hpi and the results showed that the percentage of cells harboring this [*LEF*⁺] phenotype continuously decreased from 24 to 48 hpi (see revised Fig. 5e-f).

We selected one cell from the photo, and occluded all pixels as 100%. The proportion of the pixels of a certain brightness to all the pixels is defined as "percentage of pixel". We have added new text to explain this in the legend to Figure 1c.

We had purified his-tagged LEF-10 from virus infected insect cells. Purified LEF-10 was boiled in loading buffer (with SDS and DTT) for 10 min, then we performed SDS-PAGE and Western blot. The majority of LEF-10 signal was observed in the stacking gel (Please see the image). To be consistent with other reports on aggregation proteins, we have now described LEF-10 as “aggregates”.

We have no direct evidence that the foci of LEF-10-EGFP are insoluble. However, in the revised Fig.5, we show that LEF-10-EGFP forms more foci in cells than the L21A mutant and it forms more SDS-resistant aggregates as shown by SDD-AGE and SDS-PAGE.

Line 92. Define the [PSI+] and [psi-] phenotypes.

As suggested by the reviewer, we have added a detailed definition in the revised manuscript. See Line 96-104.

Line 169-172. Alignment does not confirm a “discrete cPrD”, it only identifies potentially conserved (not “constant”) domains. Alignment is on 28 baculoviruses not “nucleopolyhedroviruses”. Alignment was previously reported by the authors and it could be referenced instead of shown and only the primary sequence of LEF10 provided.

We note this error and we have made the changes accordingly. As suggested by the reviewer, the figure of alignment has been removed and be referenced in the revised manuscript.

Line 190. Delete “Without exception”, there are only 5 constructs and 3 have the domain.

We agree with the reviewer and have now deleted “Without exception” in the revised manuscript.

Line 211. “Interestingly, the LEF-10 cPrD aligned with the most conserved region of C1” This sentence does not make sense, since the alignment is of homologs of AcMNPV LEF10?

Thanks for the comment. The alignment result showed that there are ten highly conserved amino acid residues in the C1 region, and the LEF-10 cPrD overlaps with C1 region. We have removed this sentence in the revised manuscript.

Line 211-215. Awkward run-on sentence rewrite.

As suggested by the reviewer, we have modified the sentence in the revised manuscript.

Line 215-218. The figure 6a does not show any quantitative reduction in efficiency of *ade1-14*.

We thank the reviewer for pointing this out. We have modified the relevant description and moved the figure to “Supplementary Information” in the revised manuscript.

Lines 219-226. The authors need to make it clear they are now discussing recombinant viruses and bacmid transfected insect cells. The *actin/p10* is an artificial construct which the authors are trying to use as a marker for late gene expression. Actin is a constitutive promoter that expresses genes a relatively high level. It is unknown how when combined with *p10*, if the actin promoter affects *p10* late gene regulation. Why wasn't the native *p10* promoter used?

We thank the reviewer for bringing this out. Several years ago, we started to rescue *lef-10*-null Bacmid at a time when the *lef-10* promoter region was unknown. The predicted *lef-10* promoter overlaps with the *Ac53* gene. To avoid the potential interfering effect of *Ac53*, we used a commercial plasmid pTriEx1.1, which contains a chicken β -actin promoter and a *p10* promoter for the purposes of protein expression both in mammalian cells and the baculovirus expression system. In our assay, the β -actin promoter could trigger the expression of LEF-10, which can be used to rescue the virus, and the *p10* promoter would only be activated at a late stage of infection noting that the protein expression level driven by β -actin promoter in insect cell is much lower than that driven by *p10* promoter. After transfection, the supernatant was collected and used to infect insect cells, and then the fusion proteins containing green fluorescent protein were observed using fluorescence microscope after 4 days post infection. We found that this strategy could work for LEF-10 truncations or mutants in rescue assays. If a LEF-10 mutant cannot rescue LEF-10-null bacmid, virus infection would not occur.

In this manuscript, we firstly rescued the LEF-10-null bacmid using pTriEx1.1-based transfer vector (*actin/p10* promoter), and then confirmed the results using the native *lef-10* promoter. Because we observed obvious different result between L21A and wild-type LEF-10 in the first rescue assay, we then focused on L21A mutant in the following assays. To tell the story logically, we displayed the first rescue assay using the *actin/p10* promoter in this manuscript (line 174-176 in revised edition).

It is claimed that there was significantly higher expression in L21 mutant. How was this determined and what was the quantitative assay that was used? There are no units or axis labels on the flow cytometry graphs.

We thank the reviewer for pointing this out. We used flow cytometry to quantify the expression of EGFP fusion protein. The mean fluorescence intensity of infected cells was calculated and compared (L21A:2724 vs WT:860).

As suggested by the reviewer, units and axis labels we have now added these to the flow cytometry graphs in Fig. 5b and Supplementary Figures 4a, 7b and 8b in the revised manuscript.

If there is an increase expression in the L21 mutant it could be due to increased DNA replication and copy number. This needs to be checked.

To address the reviewer's concern, we have now checked viral DNA copy number using real-time PCR and the results showed that the DNA copy number of WT and L21A mutant are similar before 24 hours post-infection, and the DNA copy number of WT is more than 2 times higher than L21A's copy number at 48 hpi. The growth curve is now shown in Supplementary Figure 5 which also shows that L21A mutation reduced the virus propagation.

In Fig. 6c there does not appear to be any significant difference, how many hours post-transfection were these samples analysed and how many repeats were performed and was the efficiency of transfection always similar and how was this confirmed? Any differences in expression could just reflect a productive infection and a functional LEF10. In Fig. 6d how many cells were analysed and at what times?

To address these questions we have now analysed infected cells at 48 hours post infection, and repeated this analysis three times. Our transfection efficiency is satisfactory. At 5 days post transfection, supernatants were collected and then used to infect *Sf9* cells. Three time independent assays were also carried out and as shown in Fig. 6, we analysed 30,000 cells for each sample.

We have previously found that after separation by SDS-PAGE, partial of LEF-10 remains in the stacking gel. We have therefore replaced Fig. 6c with a new image with the SDS-PAGE gel

containing both the stacking gel and the resolving gel and moved it to Supplementary Figure 4b in the revised manuscript.

Line 227-229. The data as shown does not support the statement that, “the phenotype of LEF-10^{L21A}-Sup35MC displayed redder color in 1/4 YPD medium than wild-type Sup35”.

According to reviewer’s comment, we have double checked the results in yeast and found that L21A mutant on YPD media displayed a slightly redder color than wild-type but the difference is not obvious.

Therefore, to further investigate if L21 is important for prion-mediated aggregation in yeast cells or not, we expressed the GFP-tagged LEF-10 and LEF-10^{L21A} in yeast cells and found that LEF-10 and LEF-10^{L21A} exhibited different non-diffuse fluorescence patterns (revised Supplementary Figure 6a). In addition, we demonstrated by SDD-AGE that both GFP-tagged LEF-10 and LEF-10^{L21A} formed similar aggregates *in vitro* (revised Supplementary Figure 6b).

However, the conclusion that L21 is important for LEF-10 aggregation has also been made from new experiments using insect cells. In infected *Sf9* cells, we observed that over-expressed LEF-10^{L21A}-EGFP formed fewer aggregates than overexpressed LEF-10-EGFP (see revised Fig. 5c). To avoid the influence of LEF-10 overexpression on virus production, we further constructed a cassette that puts LEF-10 and LEF-10^{L21A} under the control of the native *lef-10* promoter (a weak promoter). By doing so we found that infected *Sf9* cells expressing LEF-10^{L21A}-EGFP exhibited a weak [*LEF*⁺] phenotype at 24 hpi (hours post-infection), 36 hpi and 48 hpi (see revised Fig. 5e and 5f). Moreover, SDD-AGE analysis and Western blot revealed that L21A could form lower molecular-weight SDS-resistant polymers that migrated slower than the monomeric form (revised Fig. 5g and 5h).

The results allow us to conclude that L21 is associated with aggregation in infected *Sf9* cells and that the L21A mutant showed a reduced LEF-10 aggregation tendency. However, it seems that L21 is not responsible for prion-mediated aggregation in yeast cells. Since our description may cause misunderstanding, we have modified the relevant text in the revised manuscript.

Line 260. What is meant by “innate weak promoter”. How is the promoter defined as weak?

We thank the reviewer for pointing this out. We have changed “innate weak promoter” to “native weak promoter”. We define *lef-10* promoter as a “weak” promoter based on transcriptome data. The transcriptional level of *lef-10* ranks fourth from the end of all viral transcripts, and is about 1800 times less abundant than that of *p10*. (Reference: Chen YR, Zhong S, Fei Z, Hashimoto Y, Xiang JZ, Zhang S, Blissard GW. The transcriptome of the baculovirus *Autographa californica multiple nucleopolyhedrovirus* in *Trichoplusia ni* cells. *J Virol.* 2013, 87:6391-405.)

Figure 7. Statistically this experiment does not make sense. To ensure every cell gets infected an MOI of 5-10 must be used. In addition, it is assumed that significantly higher LEF10 expression will

be achieved by having a cell infected with more genomes. Is there any evidence for this to be true? The assay is a late gene promoter which is activated after DNA replication is initiated and multiple genomes are present. Again, no time post-infection is provided. The authors need to perform higher MOI infections, ensuring every cell is infected and analyze and compare multiple time points.

As suggested by the reviewer, we have now performed a new experiment using a MOI ranging 0.0625 to 64 to ensure every cell is infected. The results obtained showed that EGFP positive cells were comparable in LEF-10^{L21A} and wild-type LEF-10 groups at MOI lower than 8. However, at higher MOI, there were obviously more EGFP positive cells in LEF-10^{L21A} group than in the LEF-10 group (Fig. 6a and Supplementary Fig. 7). To address the concern about significantly higher LEF-10 expression with higher MOI, we have now performed Western blot and it showed that the expression level of LEF-10 increased with the increase of MOI from 0.0625 up to 16 (Please see the image).

As suggested by the reviewer, we have performed new infection assays at a MOI of 10 and visualized the aggregate of LEF-10 and the expression of late gene regulated by LEF-10 (Please see revised Fig 6b).

Line 298-299. Data does not show LEF10 exists in alternative conformations in insect cells.

We thank the reviewer for the comment. We have now imaged infected cells expressing the virus-encoded LEF-10 using confocal microscope and it showed that infected cells exhibited two phenotypes, reflecting the two different conformations of LEF-10 fusion proteins. In addition, Western blot (revised Fig. 5h) and SDD-AGE analysis (revised Fig. 5g) revealed that LEF-10 existed in two different conformations: monomer and aggregate. Since our description may cause misunderstanding, we have changed “alternative conformations” to “two different conformations” in the revised manuscript.

Line 316-317..” once viruses infect a cell at high multiplicity, the virus could limit its early replication to avoid the excessive stimulation of host antiviral responses” What is meant by “early replication”? What evidence is there that baculoviruses cause excessive stimulation of an antiviral response? Most antiviral responses occur rapidly upon sensing a pathogen yet the events described in this paper occur late.

We thank the reviewer for pointing out these matter and agree with the viewpoint. Therefore we have removed the relevant description and modified the text accordingly

Line 319-321. This is not necessarily true. It depends on the protein and the cell type. Authors should also read, Journal of General Virology (1990), 71, 2841-284.

We thank the reviewer for pointing out these errors. We have modified the text accordingly and cited this article in the revised manuscript.

Line 322. The authors never used a high MOI.

We have now performed a new experiment extending the MOI range from 0.125-4 (revised Supplementary Figure 8) to 0.0625-64 (revised Supplementary Figure 8) and these results suggest that LEF-10 may remain in an aggregated prion state when an insect cell is infected by multiple virions, and that the aggregation of LEF-10 will lead to the inactivation of the protein and result in the reduction of the downstream gene expression dependent on it.

Line 325. “rapid”

We thank the reviewer for pointing out this typo and we have made the change accordingly.

Reviewer #3 (Remarks to the Author):

This is an important paper with compelling evidence that the virus uses a prion protein for normal functional regulation. I suggest that the introduction should be more balanced about disease vs benefit nature of prions. While generally well written the manuscript contains a surprisingly high number of typos and grammatical errors. The native English speaking author should look carefully for these errors.

Thanks for the comment. As suggested by the Reviewer, we have amended the introduction and corrected the errors in the revised manuscript and the revised manuscript has now been carefully edited throughout by the English speaking author.

The evidence that the viral domain acts as a prion in yeast is good. One addition that would be nice is to show either non-Mmedelian segregation of cytoduction of this chimeric prion in yeast.

We agree that this would be nice - but not essential - to show, but feel that we have provided sufficient genetic and biochemical data to allow us to draw the conclusion that the LEF-10 protein can behave as a prion in yeast.

I also had trouble understanding how the virus rescue assay relates to Fig. 6B and don't see the important redder color talked about for the 21A mutant. In that regard it would be nice (but not necessary) to see in yeast that the 21A mutation prevents formation of SDS resistant oligomers.

Thanks for your suggestions. As suggested by reviewer, we have now added a diagram to clarify the rescue assay (revised Supplementary Figure 1) and have described the generation of recombinant baculoviruses in detail (Please see Methods). In brief, Bacmid Δ *lef-10* could not live in insect cells unless the gene were repaired by homologous recombination at the *polh* locus. LEF-10 mutants and LEF-10-EGFP could repair the function of LEF-10 and be used for the virus rescue and generation of recombinant baculoviruses.

For redder color talked about for the L21A mutant, we double checked the results in yeast and found that L21A mutant on YPD media displayed a slightly redder color than wild-type but the difference is not obvious. Since our description may cause misunderstanding, we have removed the relevant text in the revised manuscript.

As suggested by the reviewer, we have performed further experiments to investigate if L21 is important for prion-mediated aggregation in yeast cells or not. We expressed the GFP-tagged LEF-10 and LEF-10^{L21A} in yeast cells and found that LEF-10 and LEF-10^{L21A} exhibited different non-diffuse fluorescence patterns (revised Supplementary Figure 6a). In addition, we demonstrated by SDD-AGE that both GFP-tagged LEF-10 and LEF-10^{L21A} formed similar aggregates *in vitro* (revised Supplementary Figure 6b).

However, the conclusion that L21 is important for LEF-10 aggregation has also been made from new experiments using insect cells. In infected *Sf9* cells, we observed that over-expressed LEF-10^{L21A}-EGFP formed fewer aggregates than overexpressed LEF-10-EGFP (see revised Fig. 5c). To avoid the influence of LEF-10 overexpression on virus production, we further constructed a cassette that puts LEF-10 and LEF-10^{L21A} under the control of the native *lef-10* promoter (a weak promoter). By doing so we found that infected *Sf9* cells expressing LEF-10^{L21A}-EGFP exhibited a weak [*LEF*⁺] phenotype at 24 hpi (hours post-infection), 36 hpi and 48 hpi (see revised Fig. 5e and 5f). Moreover, SDD-AGE analysis and Western blot revealed that L21A could form lower molecular-weight SDS-resistant polymers that migrated slower than the monomeric form (revised Fig. 5g and 5h).

The results allow us to conclude that L21 is associated with aggregation in infected *Sf9* cells and that the L21A mutant showed a reduced LEF-10 aggregation tendency. However, it seems that L21 is not responsible for prion-mediated aggregation in yeast cells.

Reviewers' Comments:

Reviewer #1:

Remarks to the Author:

In large part, the authors are responsive to reviewers' comments. By dropping out the section of L21A mutant effect on [PSI⁺]-like phenotype, adding more data, and correcting some wrong references, this manuscript is significantly improved.

However, there are still some outstanding issues not addressed. For example, why there are two bands in the Western blot in Fog 4C when probed with α -Sup35C? If the authors are not able to address this, more questions will have to follow: what are the sizes of these bands, and which one is the correct one? Is it possible that the full-length Sup35 was not eliminated? If that is the case, the whole story of Sup35MC fusion is not there.

Although the authors defined [LEF⁺] and [lef⁻] as [PSI⁺]-like and [psi⁻]-like phenotypes of LEF-10-Sup35MC in yeast (line 96), later these terms were referred to as Sf9 cells with different aggregation statuses of LEF-10. This is not only confusing but also conceptually wrong as there are no genetic or biochemical data showing that these LEF-10 aggregates in Sf9 cells are infectious and thus cannot be regarded as prions.

Two minor concerns:

- 1) Line 112 – the statement that "A common feature of prion proteins is their ability to form SDS-resistant polymers" is not correct as PrP^{Sc} is SDS sensitive.
- 2) Line 124 – change "... yeast prions." to "... yeast amyloid prions." Non-amyloid prions, such as [GAR⁺] does not require Hsp104 for its propagation.

Reviewer #3:

Remarks to the Author:

The authors have answered all the concerns in my previous review. The manuscript is now well written novel report of wide interest.

Reviewer #4:

Remarks to the Author:

The authors have done a thorough job of revising the manuscript and performing new experiments to satisfy the criticisms of the reviewers. I have one comment regarding writing virus names:

The previous reviewer is correct in saying that virus species names should be written in all italics, however in this case the authors are not referring to species, but to actual physical viruses. A species is an abstract human concept, an attempt to classify a collection of related strains, and not a physical entity. Thus a virus species cannot infect anything or contain a protein. The virus names in these cases should not contain any italics at all, and none of the words in a virus name should be capitalized unless they are a proper noun. See the guidelines of the International Committee on the Taxonomy of Viruses at https://talk.ictvonline.org/files/ictv_documents/m/gen_info/7004

So the sentence on line 53 "Autographa californica multiple nucleopolyhedrovirus (AcMNPV) is a large double-stranded DNA virus that infects insects" is referring to the physical virus entity, not the species, and the name should not be italicized. If the sentence were written as "Members of the

species *Autographa californica* multiple nucleopolyhedrovirus are large double-stranded DNA viruses that infect insects”, then it would be written in italics. Similarly on line 59.

RESPONSE TO REVIEWERS' COMMENTS

NCOMMS-18-04544A: Nan *et al.* "A viral expression factor behaves as a prion"

We would like to thank all three reviewers for the time they have taken to read and comment on our revised manuscript and for their extremely helpful and perceptive comments and corrections. Below we provide detailed point-by-point responses to all the comments made by the reviewers and hope these responses will satisfy the reviewers.

Point-by-point responses to the referees' comments

Reviewer #1

... why there are two bands in the Western blot in Fig 4C when probed with α -Sup35C? If the authors are not able to address this, more questions will have to follow: what are the sizes of these bands, and which one is the correct one? Is it possible that the full-length Sup35 was not eliminated? If that is the case, the whole story of Sup35MC fusion is not there.

RESPONSE

Both we and others (see below) have routinely - but variably - observed a second lower molecular weight protein when using the Sup35C antibody in Western blot analysis of yeast extracts. This second band is widely assumed to be due to proteolytic cleavage of full-length Sup35 (76.6 kDa) that generates a stable MC-terminal fragment. In our study, the two yeast strains used to generate the blots in Fig 4b were same as in Fig 2b (see Figure below: linked with red lines) so this does not represent the impact of different genetic backgrounds. The variability was also seen when we checked our recorded repeat blots for Fig 2b where some repeats also gave two Sup35 bands (right side of the figure below). We suspect that the variable appearance of the lower mol wt protein is caused by protein degradation, in which the bigger band is the full-length protein while the smaller band is a cleaved fragment containing Sup35MC. The extent to which such cleavage is seen most likely reflects differences in sample preparation and/or storage.

NB: Theoretical molecular weight of the upper full-length LEF-10:Sup35 protein is indicated above each lane.

The truncated form seen lacks the PrD (in our case the LEF-10 sequence) and behaves as a soluble protein. For example, when a [PSI⁺] extract is separated into soluble and high mol wt pellet fraction, the full-length Sup35 is found as expected in the pellet fraction whereas the lower mol wt protein is found in the soluble fraction (data Naeimi, W.R. and Tuite, M.F. unpublished).

In addition, a number of published studies from other authors have included figures which clearly show the truncated form of Sup35. Several examples are shown below for reference:

[redacted]

A. Figure 3B in *PLoS ONE*. 2012, 7(1):e29832.

B. Figure 3E in *Cell*. 2000, 100:277–288.

C. Figure 3 in *Genetics*. 2000, 156(2):559.

D. Figure 5 in *EMBO J*. 1996, 15(12):3127-34.

To confirm that we were using strains (for the data shown in Figs 2 and 4) that were NOT expressing the full-length wild type Sup35 protein post-plasmid shuffling, all selected yeast strains were double checked by demonstrating their inability to grow on SD-Ura plate i.e. to prove they lacked the *SUP35* maintainer plasmid p[*SUP35-URA3*].

We are therefore confident that the strains we analyzed in our study (a) were not expressing wild type Sup35 from the plasmid p[*SUP35-URA3*], (b) the LEF-10-Sup35MC fusion protein truncated form of the LEF-10-Sup35MC protein was the only source of functional Sup35, and (c) the truncated form of the LEF-10-Sup35MC protein was an artefact generated by proteolysis during sample preparation.

Although the authors defined [LEF+] and [lef-] as [PSI+]-like and [psi-]-like phenotypes of LEF-10-Sup35MC in yeast (line 96), later these terms were referred to as Sf9 cells with different aggregation statuses of LEF-10. This is not only confusing but also conceptually wrong as there are no genetic or biochemical data showing that these LEF-10 aggregates in Sf9 cells are infectious and thus cannot be regarded as prions.

RESPONSE:

We agree with the reviewer that it is inappropriate to extrapolate our findings in yeast to insect

cells with regards infectious protein aggregates. Cytoplasmic genetic experiments in virus-infected insect cell remains difficult. Therefore in order to avoid confusion, [*LEF*⁺] and [*lef*] in insect cells has been replaced in the text (lines 201, 204, 205, 226 and 227) by cells with LEF^{Ag} or cells with LEF^d, respectively.

Two minor concerns:

1) Line 112 – the statement that “A common feature of prion proteins is their ability to form SDS-resistant polymers” is not correct as PrP^{Sc} is SDS sensitive.

RESPONSE:

We thank the reviewer for pointing this out and we have rephrased the sentence accordingly (line 114).

2) Line 124 – change “... yeast prions.” to “... yeast amyloid prions.” Non-amyloid prions, such as [*GAR*⁺] does not require Hsp104 for its propagation.

RESPONSE:

We thank the reviewer for pointing this out and we have rephrased the sentence accordingly (line 126).

Reviewer #3

No response required.

Reviewer #4

I have one comment regarding writing virus names: The previous reviewer is correct in saying that virus species names should be written in all italics, however in this case the authors are not referring to species, but to actual physical viruses. A species is an abstract human concept, an attempt to classify a collection of related strains, and not a physical entity. Thus a virus species cannot infect anything or contain a protein. The virus names in these cases should not contain any italics at all, and none of the words in a virus name should be capitalized unless they are a proper noun. See the guidelines of the International Committee on the Taxonomy of Viruses at https://talk.ictvonline.org/files/ictv_documents/m/gen_info/7004.

*So the sentence on line 53 “*Autographa californica multiple nucleopolyhedrovirus (AcMNPV)* is a large double-stranded DNA virus that infects insects” is referring to the physical virus entity, not the species, and the name should not be italicized. If the sentence were written as “Members of the species *Autographa californica multiple nucleopolyhedrovirus* are large double-stranded DNA viruses that infect insects”, then it would be written in italics. Similarly on line 59.*

RESPONSE:

We thank the reviewer for pointing this out. In the revised manuscript, we refer to actual physical viruses, so we changed them back into non-italics format (lines 55 and 61).

Reviewers' Comments:

Reviewer #1:

Remarks to the Author:

1) Overall, the revised manuscript is significantly improved after clearly redefining [LEF+] and [LEF-] and introducing the new terms of LEF^{Ag} and LEF^d. However, it is strongly advised not to refer the LEF-10 aggregation in Sf9 cells as prions because there are no data showing they are transmissible. There are still a couple of places in the text needed to be reworded (line 221 and 229) regarding this.

2 Although it is a weak argument by citing other published work with two bands of Sup35C to prove it is okay to do the same, the clean and distinct phenotypes of yeast cells of [LEF+] and [LEF-], as well as the fact that only one band of Sup35C was seen in Fig 2b strongly argue in favor of the authors that the two Sup35-C containing bands showing in some of the Western blots is a consequence of proteolysis during the protein sample preparation. It is less of a problem than endogenous instability of the LEF-10-Sup35MC as the proteolytic Sup35C inside the cell will produce red cells, and thus mess up with the [PSI+]-like phenotype. Such a proteolysis can be prevented by adding sufficient protease inhibitors and using better experimental procedures. The authors were certainly capable of doing it as shown in Fig 2b.

RESPONSE TO REVIEWERS' COMMENTS

NCOMMS-18-04544B: Nan *et al.* "A viral expression factor behaves as a prion"

We would like to thank the reviewer #1 for the time he (she) has taken to read and comment on our revised manuscript and for his (her) extremely helpful and perceptive comments. Below we provide detailed point-by-point responses to all the comments made by the reviewer and hope these responses will satisfy the reviewer.

Point-by-point responses to the referees' comments

Reviewer #1:

1) Overall, the revised manuscript is significantly improved after clearly redefining [LEF+] and [LEF-] and introducing the new terms of LEF^{sup} and LEF^{dsup}. However, it is strongly advised not to refer the LEF-10 aggregation in Sf9 cells as prions because there are no data showing they are transmissible. There are still a couple of places in the text needed to be reworded (line 221 and 229) regarding this.

RESPONSE:

We agree with the reviewer's suggestion and we have made the changes accordingly in the revised manuscript.

2 Although it is a weak argument by citing other published work with two bands of Sup35C to prove it is okay to do the same, the clean and distinct phenotypes of yeast cells of [LEF+] and [LEF-], as well as the fact that only one band of Sup35C was seen in Fig 2b strongly argue in favor of the authors that the two Sup35-C containing bands showing in some of the Western blots is a consequence of proteolysis during the protein sample preparation. It is less of a problem than endogenous instability of the LEF-10-Sup35MC as the proteolytic Sup35C inside the cell will produce red cells, and thus mess up with the [PSI+]-like phenotype. Such a proteolysis can be prevented by adding sufficient protease inhibitors and using better experimental procedures. The authors were certainly capable of doing it as shown in Fig 2b.

RESPONSE:

Thanks for the comments and suggestions. We will pay more attention to the effects of endogenous proteases in future researches.